# Learning Cardiac Latent Representations in Vectorcardiogram Space

**Bosong Huang**[1]   **Panzhen Zhao**[1]   **Zengxiang Li**[2,3]   **Patricia Lee**[1]   **Wei Jin**[4]
**Alan Wee-Chung Liew**[1]   **Ming Jin**[1,*]   **Shirui Pan**[1,*]

## Abstract

Electrocardiography (ECG) is a cornerstone of cardiac assessment, making the learning of informative ECG representations fundamental to tasks ranging from disease diagnosis to clinical report generation. However, existing methods operate almost exclusively in the observable ECG signal space. In practice, the standard twelve-lead ECG represents multiple projections of the same underlying cardiac electrical activity from different spatial orientations. Therefore, representation learning in the ECG space inevitably introduces substantial redundancy, which may lead to spurious correlations and increased risk of overfitting. To address this and motivated by the Frank vectorcardiogram (VCG) model, we propose learning a unified latent representation of cardiac electrical activity directly in the VCG space. We introduce LVCG, the first general self-supervised representation learning framework designed to operate in this physically grounded latent space. By learning view-invariant latent VCG representations rather than lead-specific artifacts, LVCG minimizes redundancy and improves generalization. LVCG generally outperforms ECG-space baselines across tasks, demonstrating enhanced robustness and generalization, especially in domain shift settings. Our code has been made available at https://github.com/BosonHwang/LVCG.

## 1. Introduction

Effective representation learning for Electrocardiogram (ECG) signals carries substantial and diverse physiological

---

[1]Griffith University, Australia [2]SingHealth Duke-NUS AI in Medicine Institute, Singapore [3]SingHealth AI Office, Singapore [4]Emory University, USA. Correspondence to: Ming Jin <mingjinedu@gmail.com>, Shirui Pan <s.pan@griffith.edu.au>.

*Proceedings of the $43^{rd}$ International Conference on Machine Learning*, Seoul, South Korea. PMLR 306, 2026. Copyright 2026 by the author(s).

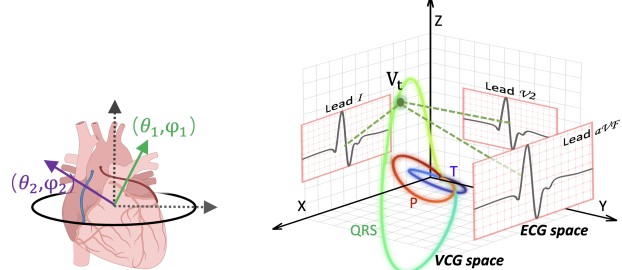

*Figure 1.* ECG and VCG as complementary views of cardiac electrical activity. (**Left**) Multi-lead ECG provides multiple linear projections of the same underlying cardiac field from different lead directions. (**Right**) Frank vectorcardiogram (VCG) offers a 3D representation of this field, and lead geometry defines a mapping between VCG space and ECG space.

and clinical significance. ECG representations are strongly associated with the detection of common cardiac disorders, such as atrial fibrillation and supraventricular tachycardia (Liu et al., 2024; Li et al., 2024). Recently, several studies have demonstrated that learning informative representations from ECG signals can facilitate the identification of previously overlooked structural heart diseases (Poterucha et al., 2025). Beyond cardiac diagnosis, ECG representations have also shown potential values, including emotion recognition (Sarkar & Etemad, 2022), blood glucose monitoring (Dave et al., 2024), and sleep apnea detection (Bahrami & Forouzanfar, 2022).

The development of ECG representation learning has explored diverse learning paradigms and modeling strategies. While supervised approaches typically focus on task specific objectives and may struggle with domain shift (Liu et al., 2024; Wang et al., 2025), self-supervised methods capture intrinsic signal characteristics to enhance robustness and generalization. From a modeling perspective, most existing methods treat multi-lead ECG data as multivariate time series (Liu et al., 2024; Baevski et al., 2020; Na et al., 2024). Despite their architectural sophistication, these methods operate entirely in the raw ECG space, where learning is hindered by two fundamental challenges.

First, there is an issue of **physical inconsistency**: by treating ECG leads as independent channels, standard models ignore the known linear dependencies imposed by the lead

field, potentially leading to physiologically inconsistent patterns that contradict the underlying cardiac electrical activity (Clifford et al., 2006). Second, there is a core issue of **feature-geometry entanglement**: the morphology of each ECG lead waveform is strongly dependent on electrode placement and individual body conductivity rather than purely on cardiac pathology (Kania et al., 2014). Without explicitly disentangling cardiac activity from acquisition geometry, waveform encoders often overfit to specific electrode configurations and generalize poorly across different subjects or devices (Jin et al., 2025).

More fundamentally, these approaches overlook the fact that ECG signals correspond to multi-view observations of the underlying 3D cardiac electrical field (Figure 1, left). Although a detailed characterization of this field is infeasible in practice (Taccardi, 1963), the Frank vectorcardiogram (VCG) (Figure 1, right) provides a compact and physically motivated abstraction (Frank, 1956). While generative frameworks like Nef-Net (Chen et al., 2021a) and its derivatives (Zhan et al., 2025; Chen et al., 2022) have utilized VCG space for lead synthesis, they focus primarily on signal reconstruction fidelity rather than feature extraction. These models lack a general representation learning paradigm for compact embeddings invariant to lead views and do not extract underlying representations that generalize beyond the generated leads themselves. Consequently, the potential of VCG space as a physically grounded bottleneck for representation learning remains largely unexplored.

To address these challenges, we propose LVCG, a framework that learns representations in a latent VCG space by identifying a unified cardiac electric field. Our approach responds to the issue of physical inconsistency by introducing a VCG space transformation which utilizes deterministic lift and project operations to enforce multi-view consistency across leads. To resolve the entanglement of cardiac activity and acquisition geometry, we employ a VCG beat encoder within a token bottleneck to extract view invariant morphology. The model follows a structured pipeline starting from beat-level processing and culminating in a temporal module that captures inter beat dynamics. This process yields a comprehensive and generalizable abstraction of the complete ECG signal, thereby establishing representation learning in latent VCG space as a promising new paradigm for ECG analysis.

Our core contributions are summarized as follows:

- We establish a general representation learning framework in the latent VCG space, offering a physically grounded alternative to direct ECG space modeling.

- We utilize a geometric transformation mechanism to ensure that the learned representations remain consistent across diverse lead configurations and electrode variability.

- We introduce a suite of modeling designs specifically for the latent VCG space, which capture the evolution of cardiac activity by integrating beat-level structure with inter beat dynamics.

- We demonstrate that LVCG achieves superior robustness and generalization across a diverse range of tasks, including linear probing classification under domain shift, multi-lead view reconstruction, and non-cardiac condition detection.

## 2. Related Work

### 2.1. ECG Representation Learning

Supervised pretraining methods have explored the use of strong labels or auxiliary modalities to learn transferable ECG representations. MERL (Liu et al., 2024) adopts supervised contrastive learning and further enhances generalization through test time clinical knowledge, enabling robust zero shot ECG classification. D-BETA (Pham et al., 2024) extends masked autoencoding to the ECG text setting and introduces additional discriminative objectives, explicitly forcing the model to distinguish reconstructed signals from original ones and leveraging diagnostic reports as strong supervision. MELP (Wang et al., 2025) performs multi-scale ECG language pretraining with contrastive alignment across token level, beat level, and rhythm level representations, improving the modeling of hierarchical temporal structure and complex arrhythmia semantics. Building on this language-aware trend, recent multimodal large language models (MLLMs) move from fixed diagnostic embeddings toward open-ended, clinically grounded interpretation. GEM (Lan et al., 2025) unifies ECG time series, 12-lead images, and text in a dual-encoder framework, linking diagnoses to waveform-level evidence such as interval measurements. TimeOmni-1 (Guan et al., 2026) formalizes time series reasoning in LLMs through tasks spanning perception, extrapolation, and decision making, with natural extensions to physiological signals such as ECG, improving causal and event-aware reasoning over temporal data.

Self-supervised approaches aim to exploit large amounts of unlabeled ECG data by designing pretext tasks that respect the spatio-temporal characteristics of multi-lead signals. ST-MEM (Na et al., 2024) employs spatio-temporal masked modeling, concatenating all leads with positional and lead embeddings and demonstrating strong linear probing and fine-tuning performance under limited labels. HeartLang (Jin et al., 2025) further interprets ECG as a language, constructing a heartbeat level codebook and representing each word with codebook entries, lead information, positional encoding, and R peak features, and learning form and rhythm representations via a spatio-temporal Transformer. Together,

these works establish powerful ECG representation learning pipelines, but all of them operate directly in the observable ECG signal space. As a result, the learned features remain sensitive to electrode placement and lead configuration, which are known to significantly affect ECG morphology and derived representations (Kania et al., 2014; Wagner et al., 2020). This lead-specific dependency limits their ability to generalize across acquisition devices and lead systems.

## 2.2. ECG related Non-cardiac Tasks

Beyond cardiac diagnosis, ECG representations have also been evaluated on non-cardiac tasks that reflect affective and physiological states. For emotion recognition, (Sarkar & Etemad, 2022) demonstrates that self-supervised pretraining on large-scale ECG corpora can significantly improve downstream affect classification, indicating that ECG encodes informative autonomic patterns beyond pathology. For sleep monitoring, (Bahrami & Forouzanfar, 2022) systematically reviews deep learning models for sleep apnea detection from ECG and shows that temporal morphology and rhythm variations are highly discriminative for respiratory disorders. These studies highlight the versatility of ECG representations and motivate learning task agnostic features that generalize across diverse semantic domains.

## 2.3. View Generation and VCG Space

The relationships among different ECG leads can be naturally interpreted from a multi-view perspective, where each lead corresponds to a projection of the underlying cardiac electrical field. This view is rooted in classical vectorcardiography. The Frank vectorcardiogram (VCG) (Frank, 1956) defines a standardized three dimensional VCG coordinate system that compactly characterizes the cardiac electric dipole and its temporal evolution. Beyond the idealized dipole assumption, physiological studies such as (Taccardi, 1963) further revealed that body surface potentials exhibit rich near-field and non-dipolar components with multiple extrema, highlighting the field-based nature of cardiac electrophysiology and motivating representations in an intrinsic electrical space rather than in individual lead signals.

Building on this physical interpretation, recent view generation methods explicitly model the mapping between different ECG leads by learning to reconstruct missing or novel views. Nef-Net (Chen et al., 2021a) demonstrates that arbitrary ECG leads can be generated from vectorcardiographic representations, while NEF-NET+ (Zhan et al., 2025) further incorporates angular information to improve cross-view synthesis and robustness. These approaches implicitly assume the existence of a latent cardiac electrical field that gives rise to multiple lead-specific observations, providing an initial computational realization of learning in the VCG space.

# 3. Methodology

We propose LVCG, a self-supervised framework that learns compact and transferable representations in a latent VCG space rather than directly in the ECG signal space. We learn view-invariant latent VCG representations via self-supervised multi-lead reconstruction. The central idea is to treat multi-lead ECGs as linear views of an underlying three-dimensional cardiac electrical field and learn representations in this latent physically grounded space. Notation is summarized in Table 6.

As illustrated in Figure 2, LVCG is organized around (1) beat-level processing, (2) VCG space transformation, (3) VCG beat encoding, and (4) a temporal module. Beat-level processing yields ECG beat patches and R–R intervals, which are encoded by the R–R Encoder into rhythm embeddings. VCG space transformation lifts visible-lead ECG patches to VCG patches and projects decoded VCG patches back to ECG leads using the lead direction matrix. The VCG beat encoder and temporal module produce structure and dynamic embeddings, and we concatenate structure, dynamic, and rhythm embeddings as the ECG representations for downstream tasks.

## 3.1. Preliminaries

**VCG Definition.** We model cardiac electrical activity as a three-dimensional trajectory $\mathbf{v}(t) \in \mathbb{R}^3$, i.e., the vectorcardiogram, VCG (Frank, 1956). Multi-lead ECGs can be viewed as different *linear views* of this underlying 3D field; each lead corresponds to a linear projection

$$e_l(t) = \mathbf{u}_l^\top \mathbf{v}(t), \tag{1}$$

where $\mathbf{u}_l \in \mathbb{R}^3$ denotes the lead direction of lead $l$. Stacking $L$ leads yields the multi-lead ECG $\mathbf{E}(t) \in \mathbb{R}^L$:

$$\mathbf{E}(t) = \mathbf{A}\mathbf{v}(t), \quad \mathbf{A} = [\mathbf{u}_1^\top; \dots; \mathbf{u}_L^\top] \in \mathbb{R}^{L \times 3}. \tag{2}$$

We refer to $\mathbf{A}$ as the lead direction matrix (or lead geometry matrix). In our implementation, $\mathbf{A}$ is fixed using direction vectors from the standard 12-lead ECG geometry (Appendix A.2); equivalently, each $\mathbf{u}_l$ can be viewed as determined by a pair of angles $(\theta_l, \phi_l)$. When the visible lead directions span $\mathbb{R}^3$ (i.e., $\mathrm{rank}(\mathbf{A}_\mathcal{I}) = 3$), the latent VCG is identifiable in the least-squares sense, enabling recovery from partial observations and projection to arbitrary target lead sets. We provide the formal identifiability statement in Appendix A.1, and the stable SVD-based recovery algorithm in Appendix A.2.

**Problem Formulation.** We formulate the ECG self-supervised learning as a *multi-lead reconstruction* problem. Given a multi-lead ECG signal $\mathbf{E} \in \mathbb{R}^{L \times T}$ with

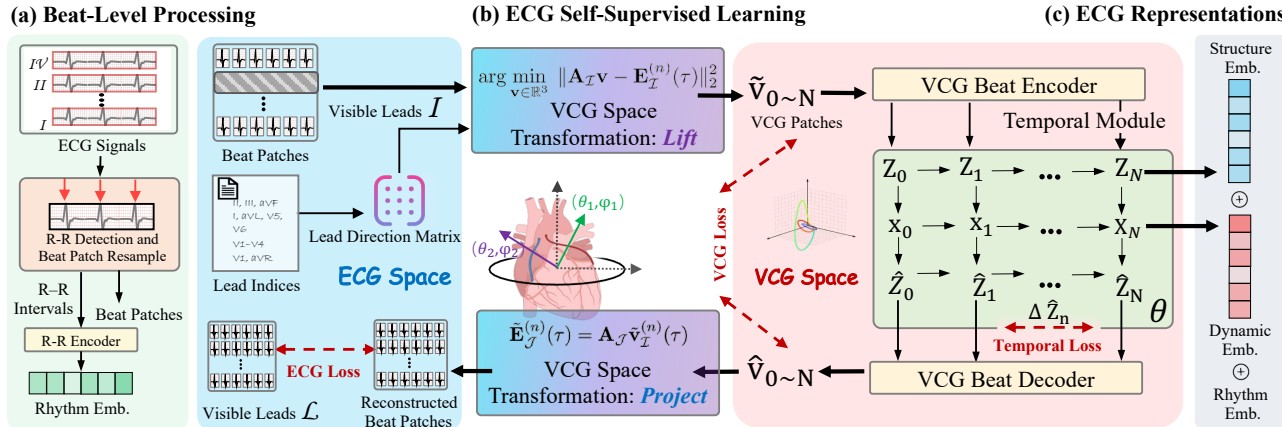

*Figure 2.* ECG self-supervised learning via multi-lead reconstruction in the LVCG framework. (**a**) Beat-level processing yields beat patches and rhythm embeddings. (**b**) VCG space transformation performs lift and project for cross-lead reconstruction. (**c**) The VCG beat encoder and temporal module produce structure and dynamic embeddings, which are concatenated with rhythm embeddings as the ECG embeddings for downstream tasks.

$L$ leads and length $T$, we sample a visible lead subset $\mathcal{I} \subseteq \{1, \ldots, L\}$ with $|\mathcal{I}| = K$ and define its complement $\mathcal{J} := \{1, \ldots, L\} \setminus \mathcal{I}$. The model is given the visible view $\mathbf{E}_{\mathcal{I}} \in \mathbb{R}^{K \times T}$ and aims to reconstruct the missing leads $\mathbf{E}_{\mathcal{J}} \in \mathbb{R}^{|\mathcal{J}| \times T}$. In practice, we pass the lead indices (e.g., visible indices for $\mathcal{I}$) to the model.

### 3.2. Latent VCG Model

**Beat-level Processing.** Given a multi-lead ECG signal $\mathbf{E} \in \mathbb{R}^{L \times T}$, LVCG extracts beat-level structure by detecting R-peaks on a reference lead and converting the continuous signal into a variable-length sequence of beat patches. Each beat patch is resampled to a fixed length $P$, yielding $\mathbf{E}^{(n)} \in \mathbb{R}^{L \times P}$ for beat index $n \in \{1, \ldots, N\}$ (stacked as $\mathbf{E}_{\text{beats}} \in \mathbb{R}^{N \times L \times P}$). We also extract R–R intervals $\mathrm{RR} \in \mathbb{R}^N$ and encode them into a global rhythm embedding $\mathbf{e}_{\text{rhythm}} = f_{\text{RR}}(\mathrm{RR})$, where $f_{\text{RR}}$ is a lightweight R–R interval encoder (Appendix A.4). The number of detected beats $N$ is data-dependent and is not assumed to be constant. Implementation details of beat segmentation, R–R extraction, and rhythm encoding are in Appendix A.4.

**VCG Space Transformation.** We define deterministic operators induced purely by lead geometry, enforcing cross-view consistency while decoupling geometric projection from representation learning. Given a visible lead subset $\mathcal{I}$ (indices of $K$ visible leads) and masked leads $\mathcal{J}$ (the complement of $\mathcal{I}$), LVCG applies a non-learnable geometry layer that recovers a VCG patch from the visible-lead ECG patch $\mathbf{E}_{\mathcal{I}}^{(n)} \in \mathbb{R}^{K \times P}$. We use $\tilde{\ }$ to denote geometry-recovered quantities, and reserve $\hat{\ }$ for decoder predictions conditioned on $\mathcal{I}$. We refer to these two operations as *lift* (ECG→VCG) and *project* (VCG→ECG). We first lift the visible-lead ECG patch to VCG space. For each within-

patch index $\tau \in \{1, \ldots, P\}$, we recover the latent field vector by a Tikhonov-regularized least-squares problem, where $\mathbf{A}_{\mathcal{I}} \in \mathbb{R}^{K \times 3}$ is the submatrix of the lead geometry matrix $\mathbf{A}$ formed by selecting rows corresponding to the visible leads $\mathcal{I}$.

$$\tilde{\mathbf{v}}_{\mathcal{I}}^{(n)}(\tau) = \arg \min_{\mathbf{v} \in \mathbb{R}^3} \ \|\mathbf{A}_{\mathcal{I}}\mathbf{v} - \mathbf{E}_{\mathcal{I}}^{(n)}(\tau)\|_2^2 + \varepsilon\|\mathbf{v}\|_2^2, \quad (3)$$

where $\varepsilon > 0$ is a Tikhonov regularization (Tikhonov, 1977) coefficient for numerical stability. We stack $\tilde{\mathbf{v}}_{\mathcal{I}}^{(n)}(\tau)$ over $\tau$ to obtain $\tilde{\mathbf{V}}_{\mathcal{I}}^{(n)} \in \mathbb{R}^{3 \times P}$. We refer to $\tilde{\mathbf{V}}_{\mathcal{I}}^{(n)}$ as a *VCG patch*, i.e., a length-$P$ window of the latent VCG trajectory in $\mathbb{R}^{3 \times P}$. In contrast to the lift operation, the project operation maps a VCG patch to any target lead set. Applied to the geometry-recovered patch, it yields the geometry-projected ECG patch:

$$\hat{\mathbf{E}}_{\mathcal{J}}^{(n)}(\tau) = \mathbf{A}_{\mathcal{J}}\hat{\mathbf{v}}_{\mathcal{I}}^{(n)}(\tau),$$
$$\text{equivalently,} \quad \hat{\mathbf{E}}_{\mathcal{J}}^{(n)} = \mathbf{A}_{\mathcal{J}}\{\hat{\mathbf{V}}_{\mathcal{I}}^{(n)}\}, \quad (4)$$

In practice we implement the lift step with a stable SVD-based solver (Appendix A.2).

**VCG Token Bottleneck Encoding.** Each geometry-recovered VCG patch $\tilde{\mathbf{V}}_{\mathcal{I}}^{(n)} \in \mathbb{R}^{3 \times P}$ (beat index $n \in \{1, \ldots, N\}$) is encoded into a VCG token

$$\mathbf{z}_n = f_{\text{enc}}(\tilde{\mathbf{V}}_{\mathcal{I}}^{(n)}) \in \mathbb{R}^D, \quad (5)$$

where $f_{\text{enc}}$ is shared across beats. The goal is a *VCG token bottleneck*: we restrict the token dimension $D$ and train the decoder to reconstruct the VCG patch from $\mathbf{z}_n$, encouraging $\mathbf{z}_n$ to retain morphology while discarding intra-patch redundancy. An information-bottleneck perspective is provided in Appendix A.5. We define a structural embedding

by mean-pooling beat tokens, i.e., $\mathbf{e}_{\text{struct}}$ is the average of $\{\mathbf{z}_n\}_{n=1}^{N}$.

Given a VCG token $\mathbf{z}_n$, the decoder predicts a VCG patch $\hat{\mathbf{V}}_{\mathcal{I}}^{(n)} = f_{\text{dec}}(\mathbf{z}_n) \in \mathbb{R}^{3 \times P}$ and projects it to ECG leads via the project operation in Eq. (4). This makes decoding explicit in the physically grounded VCG space while applying reconstruction in the ECG lead space. We perform this reconstruction per ECG patch and then concatenate the reconstructed patches along time to obtain the full-length ECG signal.

**Temporal Modeling.** Inter-beat dynamics are modeled as a latent state-space system over beat index $n$. Let $\mathbf{z}_n \in \mathbb{R}^D$ denote encoder VCG tokens and let $\mathbf{x}_n \in \mathbb{R}^d$ denote latent cardiac states. Our temporal module, parameterized by $\theta$, is defined abstractly by an initialization map $h_\theta$ from token to state, a transition $f_\theta$ for state update, and a readout $g_\theta$ from state to predicted token:

$$\mathbf{x}_1 = h_\theta(\mathbf{z}_1), \qquad \mathbf{x}_{n+1} = f_\theta(\mathbf{x}_n, \hat{\mathbf{z}}_n), \qquad \hat{\mathbf{z}}_n = g_\theta(\mathbf{x}_n). \tag{6}$$

We unroll this model autoregressively (feeding back $\hat{\mathbf{z}}_n$) to obtain a predicted token sequence $\{\hat{\mathbf{z}}_n\}$, which is then used as the decoder input to reconstruct VCG patches. We define the dynamic embedding as a summary of the trajectory, e.g., $\mathbf{e}_{\text{dyn}} := \mathbf{x}_N \in \mathbb{R}^{d_{\text{dyn}}}$. In our implementation, the temporal module is realized by a Gated Recurrent Unit (GRU) (Cho et al., 2014). Concrete instantiations of $h_\theta, f_\theta, g_\theta$ are given in Appendix A.6.

### 3.3. Learning Objectives

Training optimizes a unified objective that combines view-consistent reconstruction and temporal regularization. For each training sample, we randomly partition the multi-lead ECG into a visible lead subset $\mathcal{I}$ (size $K$) and its complement $\mathcal{J}$ (masked leads). We define token differences $\Delta \mathbf{z}_n = \mathbf{z}_n - \mathbf{z}_{n-1}$ and $\Delta \hat{\mathbf{z}}_n = \hat{\mathbf{z}}_n - \hat{\mathbf{z}}_{n-1}$, where $\hat{\mathbf{z}}_n$ is predicted by the Temporal module. We minimize

$$\mathcal{L} = \underbrace{\|\hat{\mathbf{E}}_{\theta,\mathcal{J}} - \mathbf{E}_{\mathcal{J}}\|_2^2}_{\mathcal{L}_{\text{ECG}} \text{ (ECG loss)}}$$
$$+ \lambda_{\text{vcg}} \underbrace{\sum_{n=1}^{N} \|\hat{\mathbf{V}}_{\mathcal{I}}^{(n)} - \tilde{\mathbf{V}}_{\mathcal{I}}^{(n)}\|_2^2}_{\mathcal{L}_{\text{VCG}} \text{ (VCG loss)}} \tag{7}$$
$$+ \lambda_{\text{temp}} \underbrace{\sum_{n=2}^{N} \text{Huber}(\Delta \hat{\mathbf{z}}_n, \text{stopgrad}(\Delta \mathbf{z}_n))}_{\mathcal{L}_{\text{temp}} \text{ (temporal loss)}}.$$

The first term enforces masked-lead ECG reconstruction on $\mathcal{J}$. The second term reconstructs geometry-recovered VCG patches $\tilde{\mathbf{V}}_{\mathcal{I}}^{(n)}$ from decoder predictions $\hat{\mathbf{V}}_{\mathcal{I}}^{(n)}$ conditioned

on the visible leads $\mathcal{I}$. The third term enforces temporal consistency in the token space by matching token differences. Here, $\lambda_{\text{vcg}}$ and $\lambda_{\text{temp}}$ are trade-off hyperparameters that balance these loss components. We use the Huber penalty as a robust alternative to squared error (Huber, 1992). The operator $\text{stopgrad}(\cdot)$ treats its argument as a constant in backpropagation, preventing gradients from flowing into the encoder tokens. Here $N$ is the number of beats in the signal.

In implementation, the ECG term is computed only on masked leads $\mathcal{J}$ (without zero-filling masked channels in the encoder input), the VCG term is applied only on valid beats, and the temporal term is applied only on valid beat transitions.

### 3.4. Downstream Finetuning

For downstream evaluation, we freeze the pretrained backbone of LVCG and train lightweight task-specific heads on top of the concatenated embedding (the ECG embeddings) $\mathbf{e} = [\mathbf{e}_{\text{struct}}; \mathbf{e}_{\text{dyn}}; \mathbf{e}_{\text{rhythm}}]$. We assess representation quality via linear probing and transfer learning under varying amounts of labeled data, evaluating robustness and data efficiency across multiple datasets and tasks.

## 4. Experiments

**Datasets.** LVCG is pretrained on **MIMIC-IV-ECG** (Johnson et al., 2023), a large-scale clinical 12-lead ECG collection with substantial patient and device diversity. We evaluate transfer via linear probing on the datasets reported in Table 1: **PTB-XL** (Wagner et al., 2020) with four label sets (**PTBXL-Super**, **PTBXL-Sub**, **PTBXL-Form**, **PTBXL-Rhythm**), **CPSC 2018** (also known as ICBEB) (Liu et al., 2018), and **CSN** (Chapman–Shaoxing, also known as ECG Arrhythmia) (Zheng et al., 2020). PTB-XL provides diagnostic and rhythm annotations at different granularities (e.g., coarse vs. fine diagnostic groupings and morphology and rhythm label sets), while CPSC 2018 and CSN provide arrhythmia-type annotations. Across datasets, labels are provided as multi-label indicators, with different sources adopting different taxonomies and label distributions. Lead ordering conventions can differ across corpora, so we standardize inputs via a dataset-specific lead mapping before feeding signals to the model. More detailed dataset statistics are provided in Table 9.

**Baselines.** We compare LVCG against two categories of baseline methods. **Linear Probing Classification:** We include several adapted computer vision frameworks: **Sim-CLR** (Chen et al., 2020), **BYOL** (Grill et al., 2020), **Bar-lowTwins** (Zbontar et al., 2021), **MoCo-v3** (Chen et al., 2021b), and **SimSiam** (Chen & He, 2021). For ECG-specific methods, **TS-TCC** (Eldele et al., 2021) learns ro-

*Table 1.* Linear probing results of **LVCG** and other self-supervised learning methods. The best results are **bolded**, with gray indicating the second highest. The column **Average** denotes the mean performance across all six datasets.

| Method | PTBXL-Super | | | PTBXL-Sub | | | PTBXL-Form | | | PTBXL-Rhythm | | | CPSC2018 | | | CSN | | | Average | | |
|---|---|---|---|---|---|---|---|---|---|---|---|---|---|---|---|---|---|---|---|---|---|
| | 1% | 10% | 100% | 1% | 10% | 100% | 1% | 10% | 100% | 1% | 10% | 100% | 1% | 10% | 100% | 1% | 10% | 100% | 1% | 10% | 100% |
| SimCLR (Chen et al., 2020) | 63.41 | 69.77 | 73.53 | 60.84 | 68.27 | 73.39 | 54.98 | 56.97 | 62.52 | 51.41 | 69.44 | 77.73 | 59.78 | 68.52 | 76.54 | 59.02 | 67.26 | 73.20 | 58.24 | 66.71 | 72.82 |
| BYOL (Grill et al., 2020) | 71.70 | 73.83 | 76.45 | 57.16 | 67.44 | 71.64 | 48.73 | 61.63 | 70.82 | 41.99 | 74.40 | 77.17 | 60.88 | 74.42 | 78.75 | 54.20 | 71.92 | 74.69 | 55.78 | 70.61 | 74.92 |
| BarlowTwins (Zbontar et al., 2021) | 72.87 | 75.96 | 78.41 | 62.57 | 70.84 | 74.34 | 52.12 | 60.39 | 66.14 | 50.12 | 73.54 | 77.62 | 55.12 | 72.75 | 78.39 | 60.72 | 71.64 | 77.43 | 58.92 | 70.85 | 75.39 |
| MoCo-v3 (Chen et al., 2021b) | 73.19 | 76.65 | 78.26 | 55.88 | 69.21 | 76.69 | 50.32 | 63.71 | 71.31 | 51.38 | 71.66 | 74.33 | 62.13 | 76.74 | 75.29 | 54.61 | 74.26 | 77.68 | 57.92 | 72.04 | 75.59 |
| SimSiam (Chen & He, 2021) | 73.15 | 72.70 | 75.63 | 62.52 | 69.31 | 76.38 | 55.16 | 62.91 | 71.31 | 49.30 | 69.47 | 75.92 | 58.35 | 72.89 | 75.31 | 58.25 | 68.61 | 77.41 | 59.46 | 69.32 | 75.33 |
| TS-TCC (Eldele et al., 2021) | 70.73 | 75.88 | 78.91 | 53.54 | 66.98 | 77.87 | 48.04 | 61.79 | 71.18 | 43.34 | 69.48 | 78.23 | 57.07 | 73.62 | 78.72 | 55.26 | 68.48 | 76.79 | 54.66 | 69.37 | 76.95 |
| CLOCS (Kiyasseh et al., 2021) | 68.94 | 73.36 | 76.31 | 57.94 | 72.55 | 76.24 | 51.97 | 57.96 | 72.65 | 47.19 | 71.88 | 76.31 | 59.59 | 77.78 | 77.49 | 54.38 | 71.93 | 76.13 | 56.67 | 70.91 | 75.86 |
| ASTCL (Wang et al., 2023) | 72.51 | 77.31 | 81.02 | 61.86 | 68.77 | 76.51 | 44.14 | 60.93 | 66.99 | 52.38 | 71.98 | 76.05 | 57.90 | 77.01 | 79.51 | 56.40 | 70.87 | 75.79 | 57.53 | 71.15 | 75.98 |
| CRT (Zhang et al., 2024) | 69.68 | 78.24 | 77.24 | 61.98 | 70.82 | 78.67 | 46.41 | 59.49 | 68.73 | 47.44 | 73.52 | 74.41 | 58.01 | 76.43 | 82.03 | 56.21 | 73.70 | 78.80 | 56.62 | 72.03 | 76.68 |
| ST-MEM (Na et al., 2024) | 61.12 | 66.87 | 71.36 | 54.12 | 57.86 | 63.59 | 55.71 | 59.99 | 66.07 | 51.12 | 65.44 | 74.85 | 56.69 | 63.32 | 70.39 | 59.77 | 66.87 | 71.36 | 56.42 | 63.39 | 69.60 |
| HeartLang (Jin et al., 2025) | **78.94** | **85.59** | **87.52** | 64.68 | **79.34** | **88.91** | **58.70** | **63.99** | **80.23** | 62.08 | 76.22 | **90.34** | 60.44 | 66.26 | 77.87 | 57.94 | 68.93 | 82.49 | 63.80 | 73.39 | **84.56** |
| **LVCG** | 75.33 | 79.03 | 80.13 | **70.61** | 74.62 | 79.19 | 52.28 | 59.12 | 71.24 | **72.03** | **79.87** | 83.94 | **71.09** | **79.44** | **84.15** | **62.47** | **75.17** | **84.14** | **67.30** | **74.54** | 80.47 |

bust representations via temporal and contextual contrasting on augmented ECG views. **CLOCS** (Kiyasseh et al., 2021) encourages representation similarity across spatial, temporal, and patient-level dimensions. **ASTCL** (Wang et al., 2023) integrates an adversarial module with spatio-temporal contrastive learning to enhance robustness against physiological noise. **CRT** (Zhang et al., 2024) discovers cross-domain correlations between temporal and spectral information through a cross-domain dropping-reconstruction task. **ST-MEM** (Na et al., 2024) employs spatio-temporal masked modeling to learn multi-lead ECG representations. **HeartLang** (Jin et al., 2025) treats heartbeats as words to learn form and rhythm level representations via a QRS-tokenizer and codebook. **Multi-lead Reconstruction:** For this task, we compare against: **KIM** (Kors et al., 1990), which uses multivariate regression for VCG-based reconstruction; **VAE-CNN** (Matyschik et al., 2020), employing a convolutional variational autoencoder for latent representation learning; **E-LSTM** (Sohn et al., 2020), utilizing LSTM networks to capture non-linear temporal dependencies for full ECG reconstruction; and **Nef-Net** (Chen et al., 2021a), a physically grounded generative framework for multi-lead synthesis in VCG space.

**Implementation Details.** Following standard ECG linear probing classification protocols (Jin et al., 2025), we pretrain LVCG on MIMIC-IV-ECG and then freeze the pretrained backbone. For each target dataset and task in Table 1, we train a lightweight linear classifier on the target training split, select checkpoints by validation AUC, and report test AUC. To evaluate label efficiency, we subsample the target training split to {1%, 10%, 100%} while keeping validation and test fixed. Note that several baseline results for linear probing classification are obtained from the original HeartLang paper (Jin et al., 2025). In this study, we focus on self-supervised representation learning and do not compare against supervised pre-training models such as MELP (Wang et al., 2025) or MERL (Liu et al., 2024).

For multi-lead reconstruction, we follow prior ECG view synthesis settings (Chen et al., 2021a) and define a view generation task where the model observes a subset of $K$ leads and reconstructs the remaining leads to form a full 12-lead signal. We evaluate reconstruction quality using Mean Squared Error (MSE) and Mean Absolute Error (MAE), which directly quantify waveform-level deviation from the ground truth. Results are reported under lead-availability settings $(K, 12) \in \{(3, 12), (5, 12)\}$ in Table 2.

### 4.1. Linear Probing Classification Results

Table 1 presents a comprehensive comparison of LVCG against eleven baseline methods across multiple datasets and label regimes. Our results reveal several key insights that underscore the advantages of representation learning in the latent VCG space.

The most prominent advantage of LVCG is its superior generalization across diverse clinical settings. On external datasets such as CPSC 2018 and CSN, which exhibit significant distribution shifts from the MIMIC-IV pre-training data, LVCG achieves the highest AUC across almost all label fractions. Specifically, in the extremely low-label regime (1%), LVCG outperforms HeartLang by over 10% on CPSC 2018 (71.09 vs. 60.44) and nearly 5% on CSN (62.47 vs. 57.94). By mapping multi-lead signals into a physically-grounded VCG space via a geometry-induced transformation, LVCG effectively filters out acquisition-specific artifacts such as electrode placement variability and device-specific gains.

Furthermore, LVCG demonstrates remarkable label efficiency, often achieving competitive or superior performance with only 1% or 10% of labels. For instance, on PTBXL-Rhythm, LVCG reaches best AUC of 72.03 and 79.87 at 1% and 10% respectively. The results in the 1% scenario are especially significant, as they better reflect the quality and robustness of the pre-trained representations themselves when minimal downstream supervision is available. This

*Table 2.* Results of multi-lead reconstruction performance under different lead settings on CPSC 2018 and PTB datasets.

| Methods | | CPSC 2018 | | PTB | |
|---|---|---|---|---|---|
| | | MSE↓ | MAE↓ | MSE↓ | MAE↓ |
| KIM (Kors et al., 1990) | (3, 12) | 0.85 | 0.52 | 0.74 | 0.43 |
| VAE-CNN (Matyschik et al., 2020) | (3, 12) | 0.80 | 0.49 | 1.02 | 0.56 |
| E-LSTM (Sohn et al., 2020) | (3, 12) | 0.72 | 0.48 | 0.54 | 0.38 |
| Nef-Net (Chen et al., 2021a) | (3, 12) | 0.68 | 0.40 | 1.56 | 0.52 |
| **LVCG** | (3, 12) | **0.49** | **0.37** | **0.47** | **0.36** |
| KIM (Kors et al., 1990) | (5, 12) | 1.00 | 0.47 | 1.02 | 0.75 |
| VAE-CNN (Matyschik et al., 2020) | (5, 12) | 1.02 | 0.46 | 0.99 | 0.70 |
| E-LSTM (Sohn et al., 2020) | (5, 12) | 1.02 | 0.46 | 0.94 | 0.71 |
| Nef-Net (Chen et al., 2021a) | (5, 12) | 0.44 | 0.35 | 1.00 | 0.58 |
| **LVCG** | (5, 12) | **0.40** | **0.35** | **0.78** | **0.34** |

*Table 3.* AUC scores of **LVCG** and baseline methods for non-cardiac condition detection.

| Method | Chronic Kidney Disease | | | Diabetes | | | Sepsis | | |
|---|---|---|---|---|---|---|---|---|---|
| | 1% | 10% | 100% | 1% | 10% | 100% | 1% | 10% | 100% |
| ST-MEM | 55.80 | 51.43 | 71.37 | 53.34 | 63.83 | 65.70 | 47.94 | 36.93 | 71.42 |
| HeartLang | 58.92 | 67.17 | **76.24** | 60.41 | 64.90 | 65.08 | 69.02 | 70.40 | 74.34 |
| **LVCG** | **69.04** | **73.44** | 74.66 | **63.51** | **66.28** | **67.50** | **71.68** | **74.68** | **76.01** |

efficiency is largely attributed to our VCG token bottleneck design. By compressing beat-level morphology into a single VCG token, we enforce a strong structural regularizer that prevents the model from overfitting to lead-specific noise. The superiority in rhythm-related tasks further highlights the effectiveness of our latent temporal module, which explicitly models the evolution of cardiac states in the VCG space. By disentangling cardiac dynamics from the observable multi-lead geometry, LVCG establishes a robust paradigm for universal ECG representation learning that generalizes significantly better than methods operating directly in the signal space.

### 4.2. View Reconstruction Results

Table 2 summarizes reconstruction performance under (3, 12) and (5, 12) settings on CPSC 2018 and PTB. In the information-scarce (3, 12) setting, LVCG achieves the best MSE and MAE on both datasets. On CPSC 2018, LVCG reduces MSE to 0.49 and MAE to 0.37, outperforming the strongest baseline Nef-Net (0.68/0.40). On PTB, the gap is especially pronounced: LVCG attains MSE of 0.47 and MAE of 0.36, whereas Nef-Net reaches 1.56 and 0.52, indicating that LVCG recovers missing leads with substantially lower waveform error.

Under (5, 12), LVCG achieves the lowest MSE on both datasets and the lowest MAE on PTB; on CPSC 2018 it matches Nef-Net on MAE (0.35) while improving MSE (0.40 vs. 0.44). Increasing visible leads from 3 to 5 does not uniformly reduce error: trends are dataset and metric dependent (e.g., LVCG improves on CPSC but its PTB

*Table 4.* Ablation Study on VCG Space Geometry (1% training labels).

| Method | Super | Sub | Form | Rhythm | CPSC | CSN |
|---|---|---|---|---|---|---|
| ECG Space | 51.26 | 50.34 | 45.51 | 61.11 | 50.88 | 52.53 |
| Learnable Transformation | 65.02 | 55.35 | 49.16 | 52.44 | 59.14 | 57.45 |
| Shuffle Direction Matrix | 66.54 | 58.31 | 50.44 | 62.07 | 63.12 | 57.61 |
| **LVCG** | **75.33** | **70.61** | **52.28** | **72.03** | **71.09** | **62.47** |

*Table 5.* Ablation Study on Model Components (1% training labels).

| Method | Super | Sub | Form | Rhythm | CPSC | CSN |
|---|---|---|---|---|---|---|
| w/o Bottleneck | **75.83** | 69.21 | 49.91 | 68.90 | 70.72 | 59.37 |
| w/o Temporal | 71.69 | 62.89 | **55.58** | 62.45 | 65.09 | 65.32 |
| w/o VCG Loss | 74.92 | 63.11 | 51.77 | 71.45 | 68.95 | **67.24** |
| **LVCG** | 75.33 | **70.61** | 52.28 | **72.03** | **71.09** | 62.47 |

MSE increases from 0.47 to 0.78). Together with the strong (3, 12) results and qualitative examples in Figure 3 and the Appendix (Figures 6–8), these patterns are consistent with learning a view-consistent cardiac representation in VCG space, rather than only fitting lead-specific residuals.

### 4.3. Non-cardiac Detection

We evaluate whether ECG representations transfer to non-cardiac condition detection using labels from **MIMIC-IV-ECG-Ext-ICD** (Strodthoff et al., 2024). We construct binary labels from diagnosis codes for three tasks: diabetes (ICD-10 E10–E14 and ICD-9 250), chronic kidney disease (ICD-10 N18 and ICD-9 585), and sepsis (ICD-10 A40/A41/R65.2 and ICD-9 995.9/785.5). We follow the dataset-provided folds to form disjoint train/validation/test splits. Following the linear-probing protocol, we freeze the pretrained backbone and train a lightweight classifier on top of the ECG embeddings, selecting checkpoints by validation AUC and reporting test AUC under 1%, 10%, and 100% labeled-data regimes.

Table 3 shows that LVCG consistently performs well across almost all three non-cardiac tasks. Notably, LVCG yields strong AUC in low-label regimes (1% and 10%) and remains competitive at 100%, indicating that the learned ECG embeddings transfer beyond cardiac endpoints and are robust under limited supervision.

### 4.4. Ablation Studies

We ablate key design choices in LVCG to validate the role of VCG space geometry and the representation learning components. Table 4 reports view generation ablations, and Table 5 reports ablations on model components. The corresponding full results across all label fractions are provided in Tables 12 and 13.

**VCG Space Geometry. ECG Space** removes VCG space

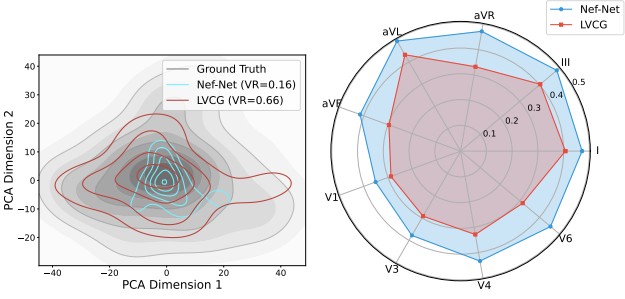

*Figure 3.* Qualitative reconstruction visualization on PTB dataset. Left: embedding-space density comparison between reconstructed leads and ground truth. Right: per-lead reconstruction errors on reconstructed leads.

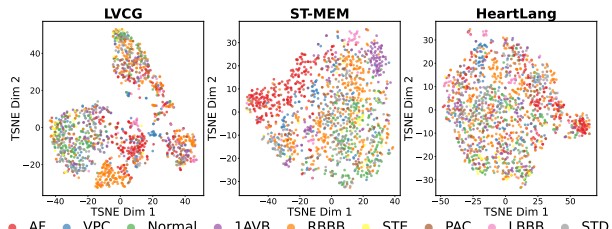

*Figure 4.* t-SNE visualization of ECG embeddings on CPSC2018 dataset. Color indicates the primary label.

transformation and performs beat encoding and decoding directly in the 12-lead ECG space. **Learnable Transformation** makes the lead direction matrix learnable and updates it end-to-end. **Shuffle Direction Matrix** permutes the lead direction matrix rows and breaks the lead–direction correspondence. All three variants show a pronounced performance drop compared to LVCG across datasets and label regimes. Together, they provide multi-view evidence that learning in VCG space requires a correct transformation and correct lead alignment. This also supports our central claim that VCG space representation learning is the foundation of LVCG, and that other modules are built on top of this physically grounded latent space.

**Model Components. w/o Bottleneck** replaces the single beat token with multiple axis-wise tokens with comparable total dimension. Table 5 shows this variant can be competitive on PTBXL-Super, but it drops on most other settings, especially on CPSC2018 and CSN. This indicates that the single-token bottleneck is an important regularizer for learning compact VCG representations that generalize across datasets. **w/o Temporal** removes the temporal module and replaces it with a simple aggregation over beats. It consistently underperforms LVCG on most tasks, highlighting the benefit of explicit beat-to-beat modeling for transferable ECG embeddings. **w/o VCG Loss** removes beat-level supervision in VCG space and optimizes the remaining objectives. Its changes are mixed, with occasional gains on some in-distribution settings but weaker cross-dataset results, suggesting that VCG loss serves as a stabilizing constraint that improves generalization.

### 4.5. Qualitative Results

We provide qualitative evidence on both multi-lead reconstruction and self-supervised learning representation quality.

**Reconstruction on PTB.** Figure 3 visualizes reconstruction behavior on the PTB dataset in the $(3, 12)$ setting where the model observes 3 input leads and reconstructs the remaining

9 leads to form a 12-lead ECG. All comparisons are performed on the reconstructed 9 leads only. Under this setting, LVCG yields reconstructions that better match the ground-truth embedding distribution and reduce errors across leads, consistent with improved view-consistent reconstruction.

**Embedding visualization on CPSC2018.** CPSC2018 exhibits a substantial distribution shift relative to MIMIC-IV pretraining data, so it provides a stronger test of cross-dataset generalization. Figure 4 suggests that LVCG produces ECG embeddings with clearer label-aligned structure, supporting transfer to downstream classification.

## 5. Conclusion

We presented LVCG, a new paradigm for ECG representation learning that moves beyond the observable ECG space to learn view-invariant latent VCG representations. By treating multi-lead ECG as multiple projections of an underlying cardiac electrical field, LVCG leverages self-supervised multi-lead reconstruction alongside beat-level bottlenecking and temporal modeling to obtain compact ECG embeddings that are decoupled from acquisition geometry. Across diverse evaluations, LVCG achieves state-of-the-art performance under label scarcity and domain shift, strong reconstruction quality, and robust transfer to non-cardiac tasks. These results suggest that learning in a latent VCG space offers a promising new perspective for general and transferable ECG representation learning.

**Limitations and Future Directions.** A primary limitation of the current framework is modeling cardiac activity as a single dipole, which serves as an approximation of the human thoracic electrical field. In reality, the potential distribution is far more complex. However, the main contribution of LVCG is demonstrating the feasibility and potential of representation learning in the VCG space. This new paradigm opens up a broad design space for future exploration, such as investigating more sophisticated cardiac field models or combining VCG representation learning with supervised pre-training. Furthermore, exploring the integration of ECG signals with LLMs within the VCG space presents an avenue for multi-modal clinical intelligence.

## Acknowledgements

S. Pan was partially supported by the Australian Research Council (ARC) under grants FT210100097 and DP240101547, and the CSIRO–National Science Foundation (US) AI Research Collaboration Program. Z. Li was partially supported by the National Medical Research Council (Singapore) under grants MOH-000655-00 and MOH-001014-00.

## Impact Statement

This work aims to improve ECG representation learning and may support future clinical decision-support systems. The model is not intended for standalone diagnosis. Potential risks include degraded performance under population shift, device-specific acquisition differences, and inappropriate use without clinical validation. Future deployment should require patient-level validation, fairness assessment across demographic groups, privacy-preserving data governance, and clinician oversight.

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

# A. Detailed Methodology

In this section, we provide additional technical details and mathematical formulations of the LVCG framework. The notations used throughout this paper are summarized in Table 6.

*Table 6.* Notation used in the paper.

| Symbol | Description |
|---|---|
| $K$ | Number of visible leads (sampled subset size) |
| $\mathbf{E} \in \mathbb{R}^{L \times T}$ | Multi-lead ECG segment with $L$ leads and length $T$ |
| $\mathcal{I}, \mathcal{J}$ | Indices of visible and masked lead subsets |
| $\mathbf{A} \in \mathbb{R}^{L \times 3}$ | Lead geometry matrix (lead direction vectors) |
| $\mathbf{A}_{\mathcal{I}}$ | Submatrix of $\mathbf{A}$ for visible leads $\mathcal{I}$ |
| $\mathbf{A}_{\mathcal{I}}^{+}$ | Tikhonov-regularized left inverse of $\mathbf{A}_{\mathcal{I}}$ (computed via SVD) |
| $\varepsilon$ | Pseudo-inverse regularization coefficient (Tikhonov) |
| $\mathbf{v}(t) \in \mathbb{R}^3$ | Latent cardiac electrical vector (VCG) at time $t$ |
| $\mathbf{V} \in \mathbb{R}^{3 \times T}$ | Latent VCG trajectory over time |
| $e_l(t)$ | ECG signal of lead $l$ at time $t$ |
| $\mathbf{u}_l \in \mathbb{R}^3$ | Direction vector of lead $l$ |
| $\mathbf{V}_{\text{beats}} \in \mathbb{R}^{N \times 3 \times P}$ | VCG patches (resampled) |
| $\mathbf{m}_{\text{beat}} \in \{0, 1\}^N$ | Beat validity mask |
| $\text{RR} \in \mathbb{R}^N$ | R–R intervals (in samples) |
| $\mathbf{z}_n \in \mathbb{R}^D$ | VCG token (beat-level representation) |
| $f_{\text{enc}}$ | Beat encoder mapping VCG patches to VCG tokens |
| $f_{\text{dec}}$ | Beat decoder mapping VCG tokens to VCG patches |
| $f_\theta$ | Latent state transition function |
| $g_\theta$ | Latent state readout / emission function |
| $\mathbf{x}_n \in \mathbb{R}^d$ | Latent state at beat index $n$ |
| $\hat{\mathbf{z}}_n$ | Predicted VCG token at beat index $n$ |
| $\Delta \mathbf{z}_n$ | Token difference $\mathbf{z}_n - \mathbf{z}_{n-1}$ |
| $\tilde{\mathbf{V}}_{\mathcal{I}}$ | Geometry-recovered VCG patch from visible leads $\mathcal{I}$ |
| $\hat{\mathbf{V}}_{\mathcal{I}}$ | Decoder-predicted VCG patch conditioned on visible leads $\mathcal{I}$ |
| $\tilde{\mathbf{E}}_{\mathcal{J}}$ | Geometry-projected ECG on target leads $\mathcal{J}$ |
| $\hat{\mathbf{E}}_{\mathcal{J}}$ | Reconstructed ECG on target leads $\mathcal{J}$ after lead projection |
| $\mathbf{e}_{\text{struct}}$ | Structural embedding (mean-pooled beat-token representation) |
| $\mathbf{e}_{\text{dyn}} \in \mathbb{R}^{d_{\text{dyn}}}$ | Dynamic embedding |
| $\mathbf{e}_{\text{rhythm}} \in \mathbb{R}^{d_{\text{rhythm}}}$ | Rhythm embedding from R–R interval encoder |
| $\mathcal{L}$ | Overall training objective |
| $\mathcal{L}_{\text{ecg-mask}}$ | Masked-lead ECG reconstruction loss |
| $\mathcal{L}_{\text{vcg}}$ | VCG patch reconstruction loss |
| $\mathcal{L}_{\text{temp}}$ | Temporal difference consistency loss |
| $D$ | VCG token dimension |
| $P$ | Resampled beat length |
| $T$ | Length of ECG signals |
| $L$ | Number of ECG leads |

## A.1. Identifiability of Latent VCG under Partial Leads

Given a visible lead subset $\mathcal{I}$ with direction matrix $\mathbf{A}_{\mathcal{I}} \in \mathbb{R}^{K \times 3}$, if $\text{rank}(\mathbf{A}_{\mathcal{I}}) = 3$, the latent VCG $\mathbf{v}(t)$ is uniquely identifiable in the least-squares sense. We denote by $\tilde{\mathbf{v}}_{\mathcal{I}}(t)$ the least-squares recovery from the visible leads:

$$\tilde{\mathbf{v}}_{\mathcal{I}}(t) = \arg\min_{\mathbf{v}} \|\mathbf{A}_{\mathcal{I}} \mathbf{v} - \mathbf{E}_{\mathcal{I}}(t)\|_2^2. \tag{8}$$

*Table 7.* Standard 12-lead direction vectors used to form the lead direction matrix $\mathbf{A}$ (MIMIC order). Each row is $\mathbf{u}_l^\top = [u_x, u_y, u_z]$.

| Lead | $u_x$ | $u_y$ | $u_z$ |
|------|-------|-------|-------|
| I    | 1.00000 | 0.00000 | 0.00000 |
| II   | 0.50000 | 0.86603 | 0.00000 |
| III  | -0.50000 | 0.86603 | 0.00000 |
| aVR  | -0.86603 | -0.50000 | 0.00000 |
| aVL  | 0.86603 | -0.50000 | 0.00000 |
| aVF  | 0.00000 | 1.00000 | 0.00000 |
| V1   | -0.33682 | 0.17365 | 0.92542 |
| V2   | 0.33682 | 0.17365 | 0.92542 |
| V3   | 0.75441 | 0.17365 | 0.63302 |
| V4   | 0.96985 | 0.17365 | 0.17101 |
| V5   | 0.92542 | 0.17365 | -0.33682 |
| V6   | 0.63302 | 0.17365 | -0.75441 |

---

**Algorithm 1** Geometry layer for VCG space transformation on a beat patch.

---

1: Form $\mathbf{A}_\mathcal{I}$ by selecting rows of $\mathbf{A}$ corresponding to visible leads $\mathcal{I}$.
2: Compute thin SVD $\mathbf{A}_\mathcal{I} = \mathbf{U}\boldsymbol{\Sigma}\mathbf{W}^\top$.
3: Construct $\mathbf{A}_\mathcal{I}^+ = \mathbf{W}\operatorname{diag}\left(\frac{\sigma_i}{\sigma_i^2+\varepsilon}\right)\mathbf{U}^\top$ (optionally threshold tiny $\sigma_i$).
4: Recover $\tilde{\mathbf{V}}_\mathcal{I} = \mathbf{A}_\mathcal{I}^+ \mathbf{E}_\mathcal{I}$ and project $\tilde{\mathbf{E}}_\mathcal{J} = \mathbf{A}_\mathcal{J}\tilde{\mathbf{V}}_\mathcal{I}$.

---

The induced view-to-view mapping is linear; in practice we use a Tikhonov-regularized left inverse $\mathbf{A}_\mathcal{I}^+$ computed via SVD (Appendix Sec. A.2):

$$\tilde{\mathbf{E}}_\mathcal{J}(t) = \mathbf{A}_\mathcal{J}\mathbf{A}_\mathcal{I}^+\mathbf{E}_\mathcal{I}(t), \tag{9}$$

which ensures consistency across arbitrary lead configurations.

## A.2. Numerical Stability of Pseudo-inverse Recovery

To avoid forming normal equations (which can amplify conditioning), we compute the regularized inverse using the thin SVD of $\mathbf{A}_\mathcal{I}$:

$$\begin{aligned}
\mathbf{A}_\mathcal{I} &= \mathbf{U}\boldsymbol{\Sigma}\mathbf{W}^\top, \\
\boldsymbol{\Sigma} &= \operatorname{diag}(\sigma_1, \sigma_2, \sigma_3), \qquad \sigma_1 \geq \sigma_2 \geq \sigma_3 \geq 0.
\end{aligned} \tag{10}$$

where $\mathbf{U} \in \mathbb{R}^{K\times 3}$ and $\mathbf{W} \in \mathbb{R}^{3\times 3}$ have orthonormal columns. The Tikhonov-regularized left inverse can be written as

$$\mathbf{A}_\mathcal{I}^+ = \mathbf{W}\operatorname{diag}\left(\frac{\sigma_1}{\sigma_1^2+\varepsilon}, \frac{\sigma_2}{\sigma_2^2+\varepsilon}, \frac{\sigma_3}{\sigma_3^2+\varepsilon}\right)\mathbf{U}^\top, \tag{11}$$

which yields the recovered field vector $\tilde{\mathbf{v}}_\mathcal{I}(t) = \mathbf{A}_\mathcal{I}^+\mathbf{E}_\mathcal{I}(t)$.

When $\mathbf{A}_\mathcal{I}$ is nearly co-planar, the smallest singular value $\sigma_3$ becomes small, making naive inversion unstable. The above form naturally suppresses directions with small $\sigma_i$ via the shrinkage factor $\sigma_i/(\sigma_i^2+\varepsilon)$. In implementation, we additionally guard against numerical issues by treating $\sigma_i$ below a tolerance (e.g., relative to $\sigma_1$) as effectively zero.

## A.3. Lead Direction Matrix from Standard 12-Lead Geometry

We implement the lead direction matrix $\mathbf{A} \in \mathbb{R}^{12\times 3}$ (summarized in Table 7) using fixed direction vectors from a standard 12-lead ECG geometry. The coordinate system is: $x$ axis right→left, $y$ axis superior→inferior, and $z$ axis anterior→posterior. Limb leads lie in the frontal plane ($z = 0$) with Einthoven-style angles, while precordial leads use an approximate transverse-plane geometry with a small inferior tilt. We use the MIMIC lead order (I, II, III, aVR, aVL, aVF, V1–V6); other datasets can be handled by reordering the rows of $\mathbf{A}$ to match the input lead order.

### A.4. Beat Segmentation and R–R Encoding

We detect R-peaks on a fixed reference lead (Lead II by default) using a lightweight peak-finding routine (SciPy). For each sample, the chosen lead is normalized (mean subtraction; optional division by standard deviation when $\sigma > 10^{-6}$), then peaks are detected with a minimum distance constraint to enforce physiological R–R bounds. If too few peaks are found, we re-run with a lower height threshold. Consecutive peaks define candidate beat intervals; we filter middle intervals by R–R length bounds and always include boundary intervals $[0, r_0)$ and $[r_{\text{last}}, T)$ to cover the full window. Each VCG interval is resampled to length $P$ via linear interpolation, yielding $\mathbf{V}_{\text{beats}} \in \mathbb{R}^{N \times 3 \times P}$, R–R intervals in samples, and a beat validity mask.

R–R intervals $\{r_n\}_{n=1}^{N}$ are encoded into a global rhythm vector $\mathbf{e}_{\text{rhythm}} \in \mathbb{R}^{d_{\text{rhythm}}}$ by aggregating: (i) global statistics (mean, std, range, CV), (ii) a sequence branch that normalizes $\log(r_n/\bar{r})$, applies a sinusoidal basis expansion, and pools with attention, and (iii) a difference branch that summarizes $\Delta r_n$ statistics. The three branches are fused by an MLP to form the final rhythm embedding.

### A.5. Beat Encoder and Information Bottleneck

Each geometry-recovered VCG patch $\tilde{\mathbf{V}}_{\mathcal{I}}^{(n)} \in \mathbb{R}^{3 \times P}$ is mapped to a VCG token $\mathbf{z}_n \in \mathbb{R}^D$ by a shared encoder

$$\mathbf{z}_n = f_{\text{enc}}(\tilde{\mathbf{V}}_{\mathcal{I}}^{(n)}). \tag{12}$$

In the main Methods section we use the information bottleneck principle to motivate this compression. Here we briefly describe the encoder architecture and give an abstract perspective on why reconstruction with a low-dimensional token behaves like a bottleneck.

The encoder $f_{\text{enc}}$ is implemented as a 1D residual network operating on the 3-channel VCG patch. Concretely, it applies a convolutional stem, followed by multiple residual stages (with temporal downsampling) to extract multi-scale morphology features, then uses global pooling over the temporal axis and a final linear projection to produce a $D$-dimensional token. The same encoder is shared across beats, so each patch is encoded independently but with shared parameters. The exact stage widths/depths and kernel sizes follow our configuration (Appendix Sec. B.1). Bottleneck behavior is induced implicitly by limiting $D$ and by shared per-beat encoding: ECG/VCG waveforms have strong local redundancy, and constraining the representation while enforcing reconstruction encourages $\mathbf{z}_n$ to retain information that is consistently useful for decoding and transfer, without requiring explicit mutual-information estimation.

### A.6. Latent State-Space Temporal Model

We model inter-beat dynamics as a latent state-space system over beat index $n$. Let $\mathbf{z}_n$ be encoder tokens and $\hat{\mathbf{z}}_n$ be predicted tokens produced by an autoregressive latent dynamics module:

$$\mathbf{x}_1 = h_\theta(\mathbf{z}_1), \qquad \hat{\mathbf{z}}_n = g_\theta(\mathbf{x}_n), \qquad \mathbf{x}_{n+1} = f_\theta(\mathbf{x}_n, \hat{\mathbf{z}}_n). \tag{13}$$

In practice, LVCG uses a deterministic causal generator without explicitly modeling process noise.

One practical instantiation sets $\mathbf{x}_n = \mathbf{h}_n$, uses $h_\theta$ as a learned initialization (e.g., a linear layer mapping $\mathbf{z}_1$ to $\mathbf{h}_1$), and implements $f_\theta, g_\theta$ with a GRU unrolled on *predicted* tokens:

$$\mathbf{h}_{n+1} = \text{GRU}(\hat{\mathbf{z}}_n, \mathbf{h}_n), \qquad \hat{\mathbf{z}}_n = \mathbf{W}_o \mathbf{h}_n, \tag{14}$$

which is autoregressive by construction. We use the final hidden state (or a pooled summary of $\{\mathbf{h}_n\}$) as the dynamic embedding $\mathbf{e}_{\text{dyn}} \in \mathbb{R}^{d_{\text{dyn}}}$.

## B. Additional Experiments Details

### B.1. Implementation Details of LVCG

The VCG token dimension is $D$ and the number of detected beats $N$ varies across segments (we optionally cap it by a configurable $N_{\text{max}}$ for memory). The BeatEncoder is a ResNet1D operating on 3-channel VCG patches and the BeatDecoder is a lightweight residual upsampling decoder. The latent dynamics module is implemented as a causal recurrent state-space

*Table 8.* Key hyperparameters of the LVCG framework.

| Category | Hyperparameter | Setting |
|---|---|---|
| Data & Preprocessing | Segment length $T$ / Sampling rate $f_s$ | 10s / 100 Hz |
| | Bandpass filter $(f_{\text{low}}, f_{\text{high}})$ | [0.67, 40] Hz |
| Beat Processing | Beat patch length $P$ / Max beats $N_{\max}$ | 128 / 20 |
| | VCG recovery regularizer $\varepsilon$ | $10^{-4}$ |
| Architecture | BeatEncoder stages / blocks / token dim $D$ | [96, 192, 256, 256] / [3, 3, 3, 2] / 256 |
| | Temporal module (GRU) hidden dim $d_{\text{dyn}}$ | 256 |
| | Rhythm embedding basis / dim $d_{\text{rhythm}}$ | 8 / 128 |
| Training | Optimizer / Batch size / Grad clip | AdamW / 64 / 1.0 |
| | Learning rate / Weight decay / Warmup | $5 \times 10^{-4}$ / 0.01 / 2000 |
| Loss Weights | $\lambda_{\text{vcg}}$ (VCG loss) / $\lambda_{\text{temp}}$ (Temporal loss) | 1.0 / 0.1 |

instantiation and produces a dynamic embedding $\mathbf{e}_{\text{dyn}} \in \mathbb{R}^{d_{\text{dyn}}}$. Pretraining samples $K$ visible leads uniformly per segment for masked reconstruction. For preprocessing, we resample to a target sampling rate $f_s$, apply a bandpass filter with cutoffs $(f_{\text{low}}, f_{\text{high}})$, and normalize each lead with per-lead z-score. Key hyperparameters are summarized in Table 8.

### B.2. Dataset Statistics

In our study, we utilize a diverse set of ECG datasets for large-scale pre-training and comprehensive evaluation. Table 9 summarizes the core statistics of these datasets. Following the protocols in HeartLang (Jin et al., 2025), we utilize the full volume of MIMIC-IV-ECG for self-supervised pre-training and evaluate the learned representations on several downstream tasks.

*Table 9.* Summary of datasets used for pre-training and evaluation.

| Dataset | Usage | # Recs | # Patients | Duration | SR (Hz) | # Leads | # Classes | Task Type |
|---|---|---|---|---|---|---|---|---|
| MIMIC-IV-ECG (Johnson et al., 2023) | Pre-training | 800,035 | 299,909 | 10s | 500 | 12 | - | SSL |
| PTB-XL (Wagner et al., 2020) | Evaluation | 21,837 | 18,885 | 10s | 500 | 12 | 5/21/44/12 | Classification |
| CPSC 2018 (Liu et al., 2018) | Evaluation | 6,877 | - | 6–60s | 500 | 12 | 9 | Cls & Recon |
| CSN (Zheng et al., 2020) | Evaluation | 10,646 | - | 10s | 500 | 12 | 11 | Classification |
| PTB (Bousseljot et al., 1995) | Reconstruction | 549 | 290 | ∼30s | 1000 | 12 | - | Reconstruction |
| MIMIC-IV-ICD (Strodthoff et al., 2024) | Non-cardiac | 800,035 | - | 10s | 500 | 12 | 3 | Binary |

**Label Granularity.** For PTB-XL, we evaluate four levels of diagnostic labels: **Super-class** (5 broad categories), **Sub-class** (21 categories), **Form** (12 morphology-related labels), and **Rhythm** (12 rhythm-related labels). This hierarchical evaluation ensures a robust assessment of our representations across different physiological characteristics.

**Non-cardiac Task.** For the non-cardiac evaluation on MIMIC-IV-ICD, we map the ICD-9/10 codes to three binary tasks: Diabetes, Chronic Kidney Disease (CKD), and Sepsis. This allows us to test the transferability of VCG-based embeddings beyond traditional cardiac diagnosis.

**Reconstruction Task.** For multi-lead reconstruction, we evaluate the performance on the PTB (Bousseljot et al., 1995) and CPSC 2018 (Liu et al., 2018) datasets. PTB provides high-resolution signals (1000 Hz) that are ideal for verifying fine-grained morphological recovery. CPSC 2018 contains signals with diverse arrhythmias and varying durations, providing a robust test for view-synthesis generalization. For both datasets, we simulate a view-generation scenario where the model reconstructs the full 12-lead ECG from a subset of $K \in \{3, 5\}$ visible leads.

## C. Additional Experimental Results

### C.1. Metrics Reflecting Morphological Preservation

Clinically meaningful ECG reconstruction quality is better characterized by morphology preservation (fiducial timing and segment level shape) rather than waveform-level amplitude deviation alone. While MSE and MAE quantify pointwise

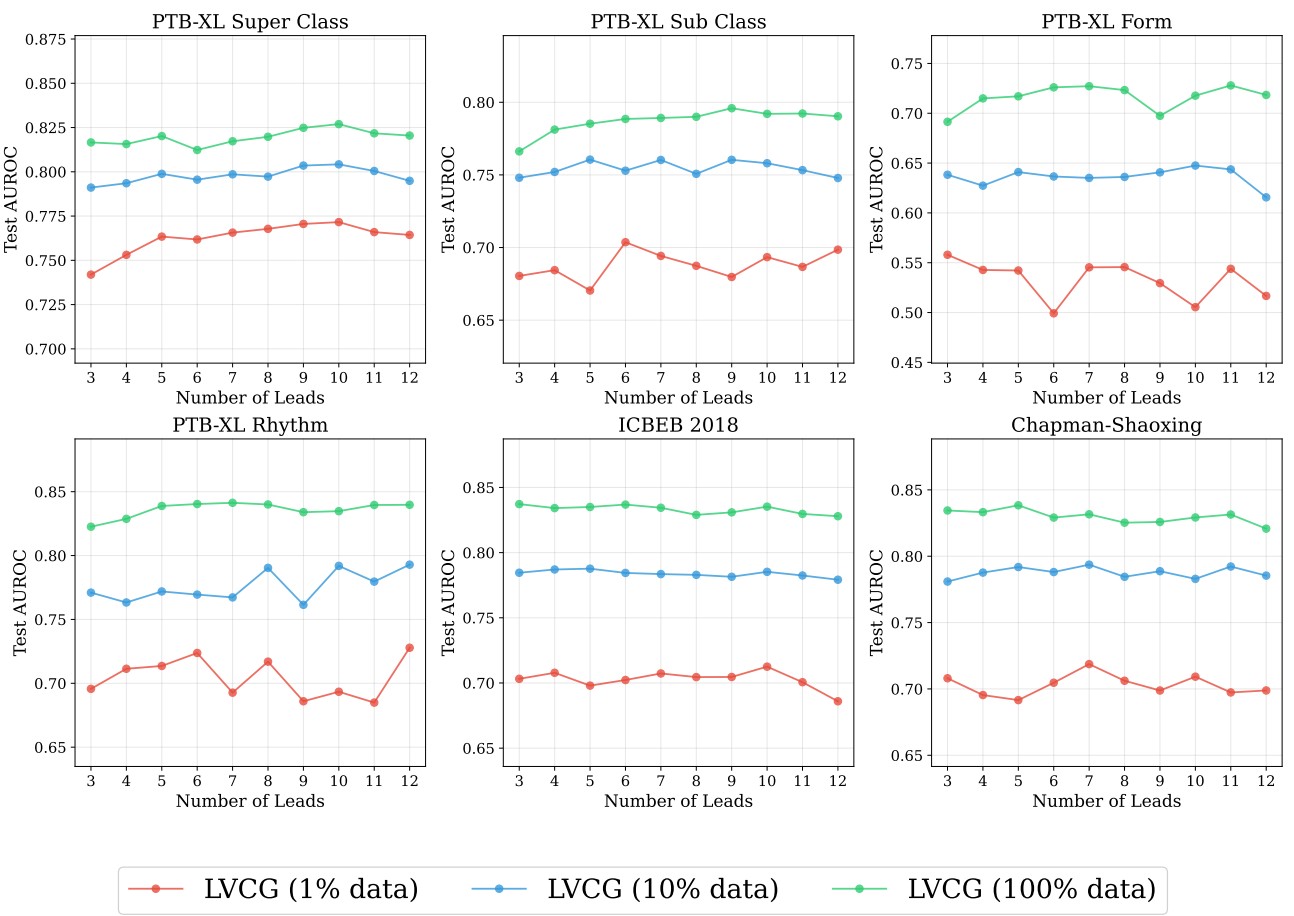

*Figure 5.* The impact study of input different numbers of leads.

*Table 10.* Reconstruction metrics reflecting morphological preservation on the PTB dataset (lower is better).

| Setting | Model | R peak err (ms)↓ | QRS dur err (ms)↓ | QT err (ms)↓ | PR err (ms)↓ | ST level err (mV)↓ |
|---|---|---|---|---|---|---|
| (3, 12) | Nef-Net (Chen et al., 2021a) | 366.21 | 33.74 | 116.27 | 22.81 | 0.23 |
| | **LVCG** | **333.39** | **23.85** | **91.71** | **20.48** | **0.20** |
| (5, 12) | Nef-Net (Chen et al., 2021a) | 782.84 | 24.63 | **88.63** | 25.22 | 0.23 |
| | **LVCG** | **374.04** | **23.94** | 91.06 | **22.81** | **0.21** |

*Table 11.* Robustness under simulated electrode perturbation (linear probing).

| Setting | AUC (%) | F1 (%) | △AUC vs. baseline (pp) |
|---|---|---|---|
| baseline | 67.55 | 27.72 | +0.00 |
| noise: 0.01 | 67.55 | 27.79 | +0.00 |
| noise: 0.05 | 67.57 | 27.86 | +0.02 |
| noise: 0.1 | 67.59 | 27.87 | +0.04 |
| noise: 0.2 | 67.41 | 27.56 | −0.14 |
| rotation: 5° | 67.57 | 27.82 | +0.02 |
| rotation: 10° | 67.56 | 27.73 | +0.01 |
| rotation: 30° | 67.39 | 27.41 | −0.16 |
| rotation: 150° | 51.73 | 13.32 | −16.82 |

reconstruction error (Table 2), they do not explicitly reflect whether standard clinical intervals and fiducials are preserved. We therefore report additional metrics on the PTB dataset that quantify errors in standard ECG features reflecting morphological preservation.

For each reconstructed 12-lead signal, we extract standard fiducials and intervals and report the absolute error relative to the reference, averaged over beats: R-peak timing, QRS duration, QT and PR intervals (ms), and ST level deviation (mV). Lower values indicate better preservation of clinically relevant morphology.

Table 10 compares LVCG against Nef-Net under $(3, 12)$ and $(5, 12)$ lead settings. In the information-scarce $(3, 12)$ setting, LVCG consistently achieves lower errors across all five measures, with particularly large gains in QRS duration (23.85 vs. 33.74 ms) and QT interval (91.71 vs. 116.27 ms), indicating more faithful recovery of clinically relevant waveform structure when only three leads are observed. Under $(5, 12)$, LVCG substantially reduces R-peak timing error (374.04 vs. 782.84 ms) while maintaining competitive QRS, PR, and ST level errors; QT error is slightly higher for LVCG (91.06 vs. 88.63 ms), suggesting that additional visible leads benefit both methods but residual repolarization timing remains challenging. Overall, these results align with the MSE/MAE trends in Table 2 and demonstrate that VCG-space reconstruction better preserves fiducial clinical features.

### C.2. Electrode Placement Robustness

Electrode placement variability can change the effective lead geometry and distort observed ECG morphology. To assess robustness under such acquisition effects, we simulate electrode perturbations at test time by modifying the fixed lead direction matrix $\mathbf{A}$ in two ways: injecting additive Gaussian noise into its entries and applying rotations of increasing angle to the lead geometry. Reported AUC and F1 are macro-averaged over the six downstream linear probing benchmarks in Table 1 (PTB-XL Super, Sub, Form, and Rhythm; CPSC 2018; and CSN) under the 1% label setting, relative to the unperturbed baseline.

Table 11 shows that LVCG maintains stable performance under moderate perturbations, with only minor changes in AUC and F1, whereas extreme rotations (e.g., 150°) lead to the expected large degradation. This pattern suggests that learning in the VCG space can tolerate moderate geometric distortions consistent with small electrode misplacement, but should not be interpreted as immunity to arbitrary lead geometry errors. We note that this study uses simulated perturbations rather than curated real world cohorts with documented electrode misplacement, validating robustness on such datasets remains an important direction for future work.

*Table 12.* Ablation Study on View Generation.

| Method | PTBXL-Super | | | PTBXL-Sub | | | PTBXL-Form | | | PTBXL-Rhythm | | | CPSC2018 | | | CSN | | |
|---|---|---|---|---|---|---|---|---|---|---|---|---|---|---|---|---|---|---|
| | 1% | 10% | 100% | 1% | 10% | 100% | 1% | 10% | 100% | 1% | 10% | 100% | 1% | 10% | 100% | 1% | 10% | 100% |
| ECG Space | 51.26 | 59.59 | 63.51 | 50.34 | 54.62 | 59.91 | 45.51 | 49.76 | 48.22 | 61.11 | 62.07 | 70.09 | 50.88 | 53.71 | 59.30 | 52.53 | 59.08 | 65.97 |
| Learnable Transformation | 65.02 | 74.08 | 77.10 | 55.35 | 61.97 | 72.87 | 49.16 | 58.29 | 66.63 | 52.44 | 61.19 | 74.80 | 59.14 | 69.26 | 76.40 | 57.45 | 68.98 | 73.54 |
| Shuffle Direction Matrix | 66.54 | 73.91 | 76.33 | 58.31 | 67.23 | 70.84 | 50.44 | 54.99 | 63.46 | 62.07 | 73.64 | 79.35 | 63.12 | 72.46 | 78.18 | 57.61 | 69.59 | 75.16 |
| **LVCG** | **75.33** | **79.03** | **80.13** | **70.61** | **74.62** | **79.19** | **52.28** | **59.12** | **71.24** | **72.03** | **79.87** | **83.94** | **71.09** | **79.44** | **84.15** | **62.47** | **75.17** | **84.14** |

*Table 13.* Ablation Study on Model Components.

| Method | PTBXL-Super | | | PTBXL-Sub | | | PTBXL-Form | | | PTBXL-Rhythm | | | CPSC2018 | | | CSN | | |
|---|---|---|---|---|---|---|---|---|---|---|---|---|---|---|---|---|---|---|
| | 1% | 10% | 100% | 1% | 10% | 100% | 1% | 10% | 100% | 1% | 10% | 100% | 1% | 10% | 100% | 1% | 10% | 100% |
| w/o Bottleneck | **75.83** | 79.52 | 81.78 | 69.21 | 73.52 | 77.22 | 49.91 | 58.18 | 70.59 | 68.90 | 74.98 | 83.87 | 70.72 | 77.52 | 83.00 | 59.37 | 68.31 | 82.43 |
| w/o Temporal | 71.69 | 77.67 | 80.54 | 62.89 | 71.87 | 77.71 | **55.58** | 60.07 | 66.96 | 62.45 | 72.73 | 80.67 | 65.09 | 74.58 | 82.11 | 65.32 | 74.47 | 80.67 |
| w/o VCG Loss | 74.92 | **79.74** | **82.14** | 63.11 | 72.61 | 77.83 | 51.77 | **60.46** | **71.74** | 71.45 | 78.60 | **85.37** | 68.95 | 77.42 | 83.50 | **67.24** | **76.91** | 82.44 |
| **LVCG** | 75.33 | 79.03 | 80.13 | **70.61** | **74.62** | **79.19** | 52.28 | 59.12 | 71.24 | **72.03** | **79.87** | 83.94 | **71.09** | **79.44** | **84.15** | 62.47 | 75.17 | **84.14** |

## C.3. Linear Probing Results under Label Scarcity and Lead Sparsity

We further investigate the robustness of LVCG by varying both the available training labels (1%, 10%, and 100%) and the number of input leads at test time (from 3 to 12). Figure 5 illustrates the performance across six datasets and tasks.

The most surprising observation is the remarkable stability of LVCG across different lead configurations. Across all tasks and label regimes, the AUROC curves remain nearly flat as the number of input leads decreases from 12 down to 3. This provides strong empirical validation for our latent VCG formulation: as long as the visible leads provide a sufficient spatial span to recover the underlying cardiac electrical field (i.e., at least 3 leads), LVCG can effectively map the signals into a consistent latent space for downstream classification.

Furthermore, LVCG demonstrates high **label efficiency** regardless of lead sparsity. The performance gap between 10% and 100% labeled data is relatively small across all datasets, and even with only 1% of labels, LVCG maintains competitive AUROC scores (e.g., above 0.70 on PTB-XL Super-class and ICBEB 2018). This capability is particularly advantageous for real-world clinical applications where both high-quality annotations and full 12-lead acquisitions may be limited, such as in wearable patch-type monitoring or emergency scenarios.

## C.4. Detailed Ablation Study Results

We provide full evaluation results for our ablation studies across all datasets and label regimes.

**VCG Space Geometry.** Table 12 compares LVCG against variants using direct ECG space learning, learnable transformations, or shuffled geometry matrices. The significant performance drop in the ECG Space baseline (e.g., from 75.33 to 51.26 AUROC on Super-class at 1%) underscores that the fixed, physically-grounded VCG transformation provides a crucial inductive bias that simpler data-driven mappings fail to capture.

**Model Components.** Table 13 breaks down the contribution of the token bottleneck, temporal module, and VCG reconstruction loss. The **w/o Temporal** variant shows a noticeable decline in rhythm-related tasks, while the **w/o Bottleneck** version exhibits reduced generalization under extreme label scarcity (1%). LVCG maintains the most robust performance across all clinical endpoints by synergizing these components.

## C.5. Reconstruction Visualization

We provide qualitative comparisons of multi-lead reconstruction between LVCG and Nef-Net (Chen et al., 2021a) to illustrate the differences in signal fidelity. Figures 6–8 showcase reconstruction cases from the PTB dataset under the (3, 12) configuration, where the model observes 3 leads (VIS) and reconstructs the remaining 9 (MASK).

As shown in Figure 6, Nef-Net (MSE: 0.6863) struggles to recover the true morphology of the masked leads, often producing attenuated or flat signals that fail to capture essential P-wave, QRS, and T-wave characteristics. In contrast, LVCG (MSE: 0.3422) demonstrates significantly higher fidelity, with the reconstructed waveforms (red) closely tracing the ground truth (gray) across all leads. This qualitative difference highlights that by learning in the latent VCG space, LVCG captures the underlying 3D cardiac electrical source more effectively than methods that lack a strong physical inductive bias. The visual consistency with ground truth supports our quantitative findings that lower MSE/MAE scores correspond to superior morphological recovery in the signal space.

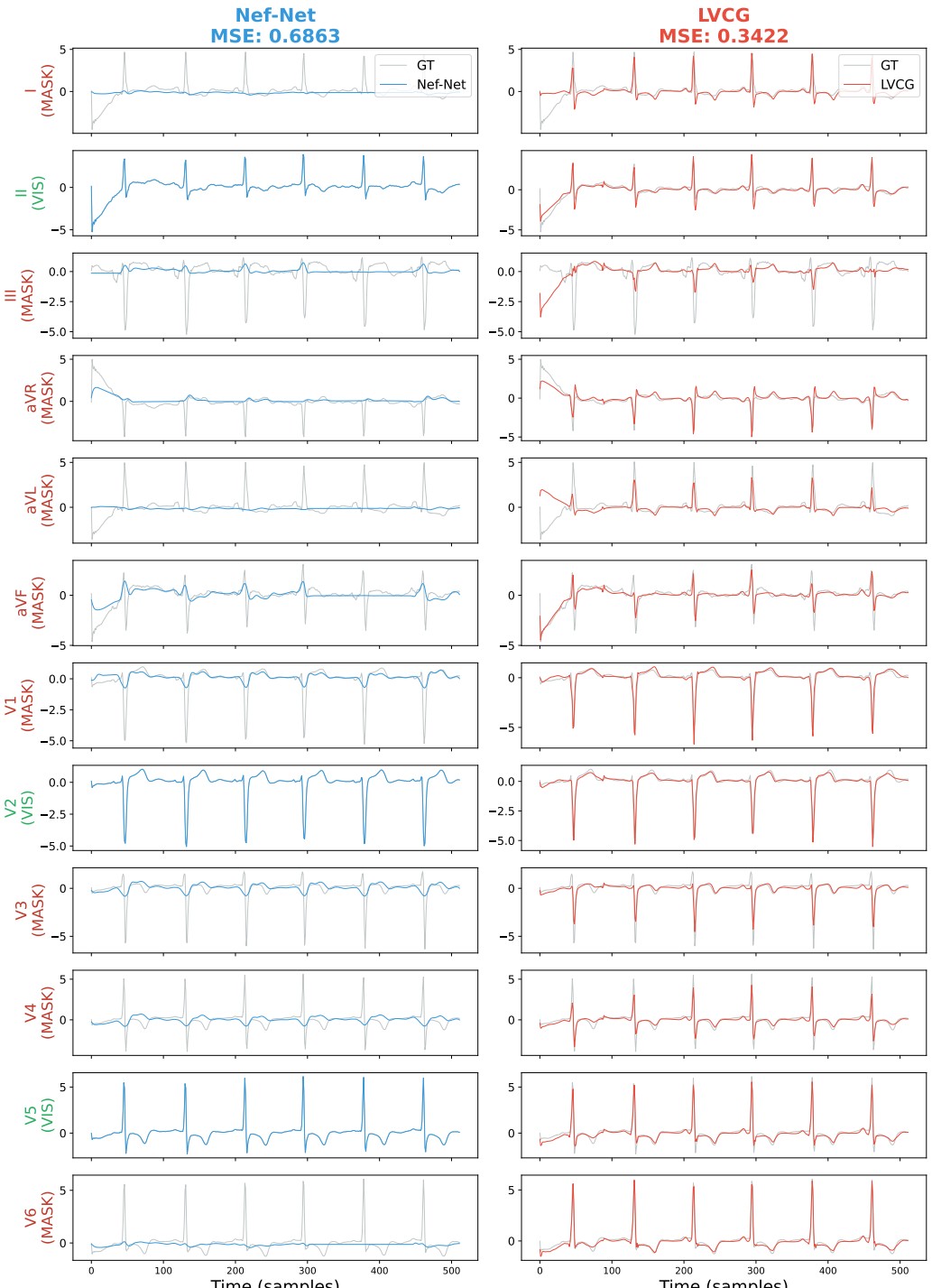

*Figure 6.* Reconstruction visualization on sample s0012_re from PTB dataset.

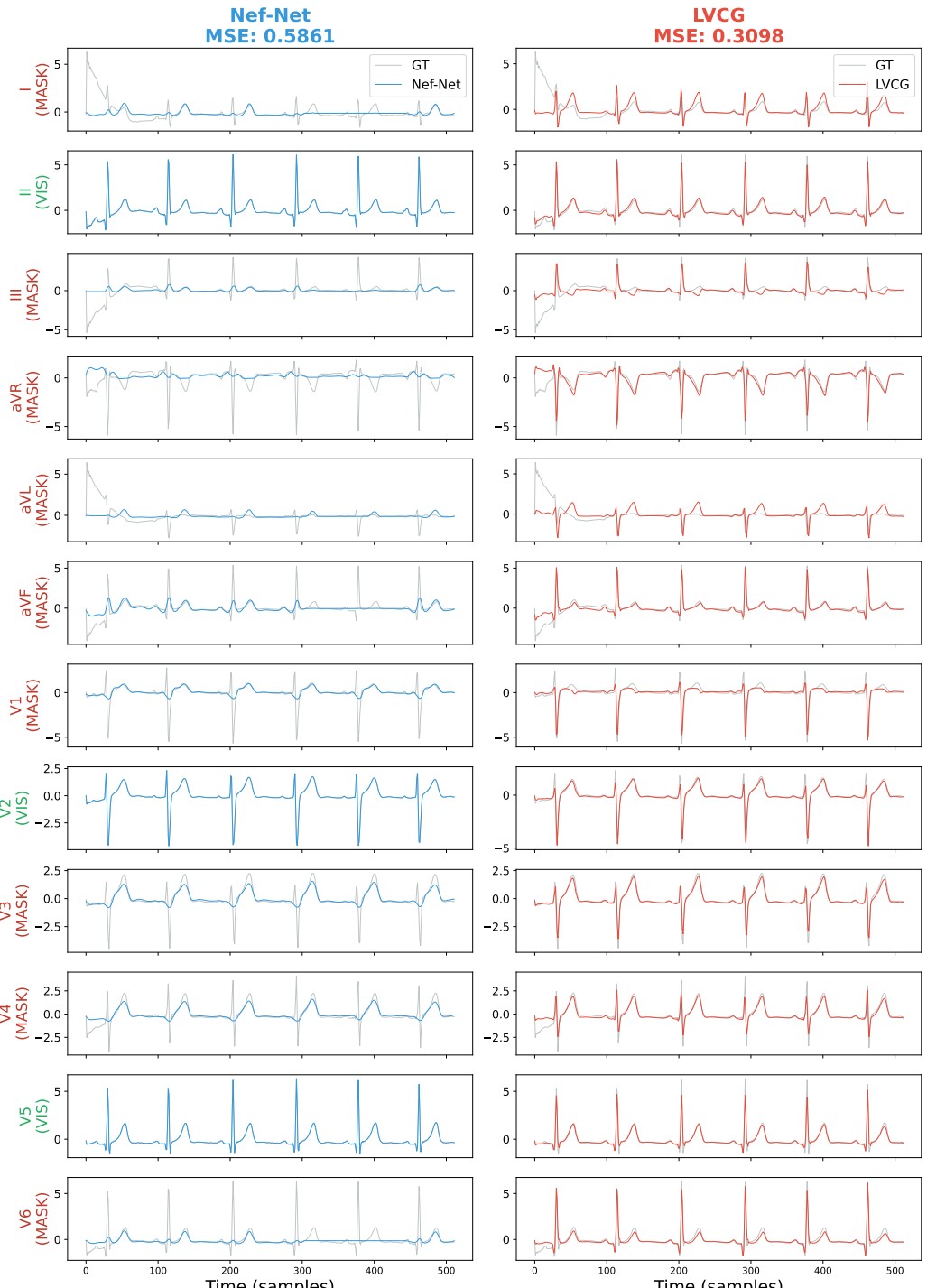

*Figure 7.* Reconstruction visualization on sample s055_re from PTB dataset.

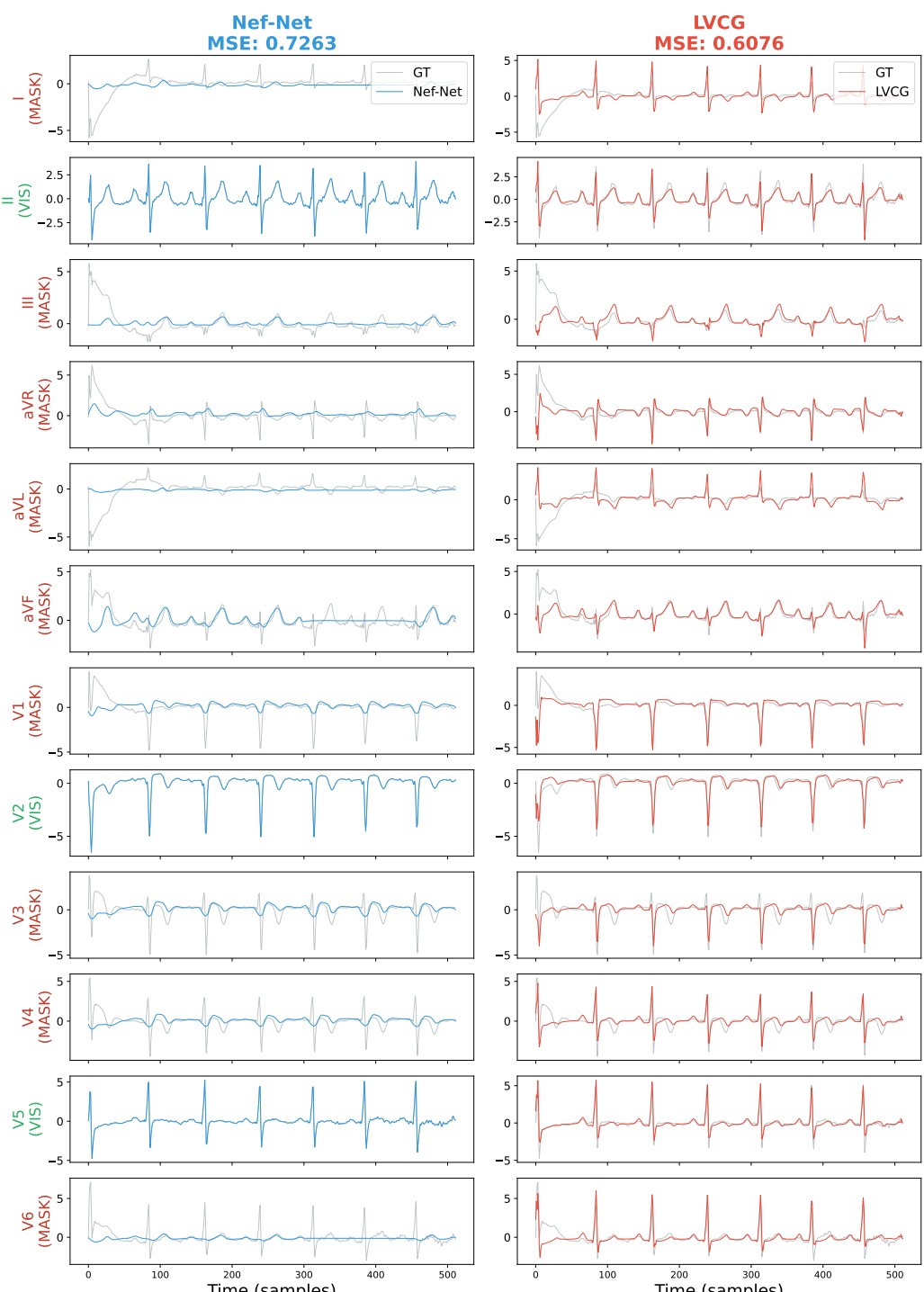

*Figure 8.* Reconstruction visualization on sample s0164lre from PTB dataset.

