# OpenReview forum: "Learning Cardiac Latent Representations in Vectorcardiogram Space"
_ICML.cc/2026/Conference — ICML 2026 regular_

### Official Review · Reviewer_paWM · 2026-03-04

**Soundness:** 3
**Presentation:** 3
**Significance:** 2
**Originality:** 3
**Overall Recommendation:** 4
**Confidence:** 3

**Summary:**

This paper introduces LVCG, a self-supervised representation learning framework that shifts away from traditional signal-space modeling by learning cardiac features directly in a physically grounded latent Vectorcardiogram (VCG) space. By treating multi-lead ECGs as linear projections of an underlying 3D cardiac electrical field, the authors utilize a geometric transformation mechanism—consisting of lift and project operations—to disentangle intrinsic cardiac pathology from acquisition-specific artifacts like electrode placement and device variability. The framework integrates a beat-level token bottleneck to extract view-invariant morphology and a latent state-space temporal module to capture inter-beat dynamics, ultimately producing compact embeddings that significantly outperform existing ECG-space baselines in tasks involving label scarcity, domain shift, and non-cardiac condition detection

**Compliance With Llm Reviewing Policy:**

Affirmed.

**Key Questions For Authors:**

1. **Robustness to Anatomical and Lead Placement Variability**
LVCG relies on a fixed, standard 12-lead geometry matrix ($A$) to perform the Lift operation. In real-world clinical settings, electrode placement often deviates from the standard, and patient torso anatomy varies significantly. Does the framework include mechanisms to account for this type of “geometric noise,” or have the authors considered introducing a learnable or personalized transformation layer to handle electrode placement deviations and anatomical variability? This would help evaluate whether the model remains robust in practical clinical environments such as emergency care or wearable monitoring scenarios.

2. **Potential Information Loss in the Token Bottleneck**
The VCG token bottleneck is designed to remove intra-patch redundancy by compressing each beat into a single token. While this design may improve generalization, could such aggressive compression risk losing subtle morphological features, such as small ST-segment deviations or minor T-wave inversions, which are often clinically important for detecting early-stage ischemia or rare cardiac conditions? Clarifying this trade-off would help determine whether the compression strategy is suitable for high-precision diagnostic tasks.

3. **Disentangling the Contributions of VCG Space vs. Temporal Modeling**
The results demonstrate improved robustness under domain shift. Could the authors further quantify how much of this improvement comes from the physical consistency constraints in VCG space versus the GRU-based temporal modeling of inter-beat dynamics? Understanding the relative contribution of these components would help clarify whether the primary advantage of the framework lies in its spatial-geometric representation or its temporal rhythm modeling capability.

**Limitations:**

yes

**Strengths And Weaknesses:**

**Strengths**

+ Physics-Driven Disentangled Modeling: The framework innovatively shifts from traditional signal-space modeling to a 3D latent space using the Frank VCG model. This design effectively disentangles intrinsic cardiac electrical activity from acquisition-specific artifacts like electrode placement and body conductivity, thereby reducing feature redundancy.
+ Superior Generalization and Label Efficiency: By utilizing a VCG token bottleneck and self-supervised reconstruction, LVCG demonstrates remarkable robustness against domain shifts and extreme label scarcity (e.g., using only 1% of labels). Experiments show that performance remains highly stable even when the number of input leads is reduced from 12 to 3.

**Weaknesses**

+ Simplified Dipole Assumption: The framework approximates cardiac activity as a single dipole, which is a simplification of the actual, more complex human thoracic electrical field. In reality, surface potentials contain rich near-field and non-dipolar components that this model may not fully capture
+ Dependency on Lead Geometry Matrix: The Lift operation relies heavily on a fixed, predefined lead direction matrix ($A$). In clinical practice, individual anatomical variations or non-standard electrode placement could lead to mismatches between actual projections and the fixed matrix used by the model, potentially introducing errors.
+ Increased Computational Complexity: Compared to direct signal processing, LVCG introduces additional preprocessing steps, including R-peak detection, beat segmentation, resampling, and SVD-based pseudo-inverse calculations. These mathematical transformations and the management of multi-branch embeddings (structure, dynamics, and rhythm) increase the computational overhead of the pipeline.

---

> ### Author Rebuttal · Authors · 2026-03-31
>
> ## Rebuttal to Reviewer paWM
>
> ### W1: Single-dipole assumption
> We thank the reviewer for raising this important concern. While we briefly discussed this limitation in the original manuscript, we agree that the analysis was not sufficiently detailed.
>
> **The single-dipole assumption may become less accurate in non-dipolar pathological conditions**, such as complex conduction abnormalities or multi-source electrical activity. However, LVCG is designed to learn a structured latent representation that captures dominant cardiac dynamics rather than perfectly modeling the full electrophysiological field.
>
> **Empirical evidence suggests that this inductive bias remains effective across diverse conditions**, indicating that the VCG space serves as a useful approximation rather than a strict physical constraint. We will expand this discussion in the revised version.
>
> ---
>
> ### W2: Dependency on lead geometry matrix
> Thank you for this insightful question. **Please see our response to Q1 below (Table R8)**.
>
> ---
>
>
> ### W3: Computational complexity
> We appreciate this concern and provide detailed runtime analysis below.
>
> **Despite additional preprocessing steps, LVCG achieves significantly lower latency compared to HeartLang.**
>
> ### Table R6: End-to-End Latency
> | Setting | Model | Mean (ms) | P50 (ms) | P95 (ms) |
> |---|---:|---:|---:|---:|
> | Batch=8 | LVCG | 32.056 | 36.979 | 39.202 |
> | Batch=8 | HeartLang | 146.945 | 143.045 | 162.670 |
>
> **Table R6 shows that LVCG is approximately 4–5× faster than HeartLang in end-to-end inference.**
>
> ### Table R7: LVCG Stage Breakdown (Inference, Batch=8)
> | Stage | Mean (ms) | P50 (ms) | P95 (ms) |
> |---|---:|---:|---:|
> | resample | 0.051 | 0.048 | 0.067 |
> | vcg transformation | 0.304 | 0.284 | 0.450 |
> | beat segment | 6.408 | 5.985 | 9.535 |
> | beat encode | 2.194 | 2.098 | 3.202 |
> | temporal modeling | 1.920 | 1.797 | 2.923 |
> | rr_embed | 1.313 | 1.266 | 1.876 |
>
> **Table R7 shows that the additional VCG transformation introduces negligible overhead**, confirming that the overall pipeline remains computationally efficient.
>
> ---
>
> ### Q1: Electrode variability
> We thank the reviewer for raising this concern. To investigate robustness to electrode placement variability, we conduct a simulation study in two ways: by rotating the lead geometry and by injecting noise into it.
>
> ### Table R8: Robustness under Electrode Perturbation
> | Setting | AUC (%) | F1 (%) | ΔAUC vs baseline (pp) |
> |---|---:|---:|---:|
> | baseline | 67.55 | 27.72 | +0.00 |
> | noise:0.01 | 67.55 | 27.79 | +0.00 |
> | noise:0.05 | 67.57 | 27.86 | +0.02 |
> | noise:0.1  | 67.59 | 27.87 | +0.04 |
> | noise:0.2  | 67.41 | 27.56 | -0.14 |
> | rotation:5   | 67.57 | 27.82 | +0.02 |
> | rotation:10  | 67.56 | 27.73 | +0.01 |
> | rotation:30  | 67.39 | 27.41 | -0.16 |
> | rotation:150 | 51.73 | 13.32 | -16.82 |
>
>
> **Table R8 shows that LVCG maintains stable performance under moderate perturbations, with only minor degradation, while extreme perturbations lead to expected performance drops.**
> **This suggests that learning in VCG space provides robustness to moderate geometric distortions caused by electrode misplacement.**
>
> We acknowledge that this evaluation is based on simulated perturbations rather than real-world datasets with electrode misplacement. **We will revise our claims to be more precise and avoid overstating robustness**, and highlight this as an important direction for future work.
>
>
>
> ---
>
> ### Q2: Preservation of clinically relevant features in bottleneck design
> Thank you for this important question. To directly assess potential information loss, **we evaluate morphology preservation focusing on clinically critical features such as ST-segment levels**, which are essential for many diagnostic tasks.
>
> **As shown in Table R5, LVCG preserves these clinically important features with comparable or improved accuracy**, indicating that the bottleneck does not remove diagnostically relevant information.
>
> While the bottleneck introduces a trade-off between invariance and fine-grained detail, **our results suggest that it primarily removes redundant variations rather than clinically meaningful signal components.**
>
> We will further clarify this trade-off in the revised manuscript.
>
> ### Q3: Disentangling the contributions of VCG space vs. temporal modeling
>
> We thank the reviewer for this insightful question. We disentangle the two components using two complementary experiments: robustness under electrode perturbation (Table R8), which mainly probes the VCG-based spatial representation, and temporal ablation (Table R3), which isolates inter-beat modeling. The former shows stable performance under moderate geometric distortion, while the latter shows that the autoregressive formulation consistently outperforms teacher-forcing variants. **Together, these results suggest that the VCG-based representation is the primary source of robustness, while temporal modeling provides additional performance gains.**

---

> > ### Author Rebuttal · Reviewer_paWM · 2026-04-01
> >
> > The author's response addressed some of my concerns. I will maintain my positive score of 4.

---

### Official Review · Reviewer_MkJU · 2026-03-10

**Soundness:** 3
**Presentation:** 3
**Significance:** 3
**Originality:** 4
**Overall Recommendation:** 5
**Confidence:** 5

**Summary:**

This paper proposes LVCG, a self-supervised ECG representation learning framework that operates in a latent vectorcardiogram space rather than the observable ECG signal space. The core idea of using the Frank VCG model as a physically grounded bottleneck for learning view-invariant representations is well-motivated and genuinely novel. The experimental evaluation is comprehensive and the ablation studies are systematic.

**Compliance With Llm Reviewing Policy:**

Affirmed.

**Final Justification:**

Thank you for the thorough rebuttal. I am satisfied that all concerns have been adequately addressed and maintain my original score.

**Key Questions For Authors:**

see above

**Limitations:**

see above

**Strengths And Weaknesses:**

Strength

- LVCG introduces a physically principled inductive bias by grounding representation learning in the VCG space, offering a conceptually clean and well-motivated departure from standard ECG-space methods.

- The fixed lead direction matrix and SVD-based lift-project mechanism provide a non-learnable geometric prior that is directly interpretable, in contrast to purely data-driven approaches.

- The multi-task evaluation covering linear probing, multi-lead reconstruction, and non-cardiac condition detection is thorough, and cross-dataset generalization results are compelling.

Weakness

- The single dipole assumption underlying the Frank VCG model is not adequately analyzed in the context of non-dipolar pathologies. For conditions such as bundle branch blocks or myocardial infarction that are known to produce non-dipolar electrical activity, the validity of the VCG prior as a representation bottleneck is unclear. The paper acknowledges this limitation but does not provide any empirical or theoretical analysis of when it may fail.

- The validity of the fixed lead direction matrix A is not robustly established. Real-world electrode placement variability and inter-subject conductivity differences cause true lead directions to deviate from the standardized vectors in Table 7. The "Shuffle Direction Matrix" ablation variant achieves 66.54 AUC on PTB-XL Super at 1%, compared to LVCG's 75.33, but the gap is not as large as might be expected if precise geometry were truly essential. This raises the question of whether the model is genuinely exploiting the physical prior or simply benefiting from a structured regularization effect.

- The comparison with HeartLang in Table 1 lacks experimental control. The paper explicitly notes that several HeartLang baseline results are taken directly from the original paper, with no guarantee of matched pretraining data volume, model scale, or hyperparameter configuration. HeartLang outperforms LVCG on PTB-XL across all full-label settings, and this gap is not discussed. Direct comparison without controlled re-implementation overstates the relative advantage of LVCG in these settings.

- The non-cardiac evaluation design introduces a potential confound. Both pretraining (MIMIC-IV-ECG) and the non-cardiac evaluation labels (MIMIC-IV-ICD) are drawn from the same patient population. This means the observed transfer performance may partially reflect within-cohort statistical associations rather than genuine representation generalizability across domains.

- The ablation in Table 5 reveals that "w/o Bottleneck" outperforms LVCG on PTBXL-Super (75.83 vs 75.33 at 1%) and is competitive on several other settings. The paper attributes this to reduced cross-dataset generalization, but does not provide a principled explanation for why the single-token bottleneck is sometimes a liability on in-distribution tasks. A more rigorous analysis of this tradeoff is needed.

- The PSNR inconsistency in Table 2 is not adequately addressed. In the (3,12) PTB setting, Nef-Net achieves higher PSNR (22.12) than LVCG (21.77), and the same pattern holds in the (5,12) CPSC setting. The paper argues that PSNR is a poor metric for 1D signals and that MSE/MAE are more informative, but this argument is applied selectively to favor LVCG. A more principled discussion of metric choice is needed, ideally supported by a clinical evaluation of morphological recovery quality.

---

> ### Author Rebuttal · Authors · 2026-03-31
>
> ### W1: Limitation of the single-dipole assumption
> We thank the reviewer for this insightful question. The single-dipole assumption underlying the VCG model may become less accurate in **non-dipolar cardiac conditions**, such as bundle branch blocks, myocardial infarction, or complex conduction abnormalities, where electrical activity may involve multiple or localized sources not fully captured by a single global dipole. However, **LVCG does not aim to perfectly reconstruct the full electrophysiological field**, but rather to provide a compact and structured latent space that captures the dominant global dynamics.
>
> **Empirically, our results suggest that this inductive bias remains effective even in diverse pathological settings**, indicating that the VCG space serves as a useful approximation rather than a strict physical constraint. We will clarify this limitation and its implications in the revised manuscript.
>
> ---
>
> ### W2: Electrode placement variability
> Thank you for this question. **Robustness under simulated electrode perturbations is reported in our response to Reviewer paWM (Q1; Table R8).** Here we complement that analysis with reduced-lead settings (e.g., limb-only and chest-only configurations), which simulate realistic variations in lead availability.
>
> ### Table R4: Lead-Subset Robustness (AUROC @ 1% Label)
> | Variant | PTBXL-Super | PTBXL-Sub | PTBXL-Form | PTBXL-Rhythm | ICBEB | Chapman | Overall |
> |---|---:|---:|---:|---:|---:|---:|---:|
> | LVCG | 76.17 | 69.88 | 51.60 | 73.38 | 71.20 | 64.06 | 67.72 |
> | limb_only | 73.00 | 56.38 | 47.42 | 64.42 | 68.13 | 60.26 | 61.60 |
> | chest_only | 58.45 | 57.31 | 53.50 | 53.79 | 55.00 | 56.91 | 55.83 |
> | Δ (limb_only) | -3.17 | -13.50 | -4.18 | -8.96 | -3.07 | -3.80 | -6.11 |
> | Δ (chest_only) | -17.72 | -12.57 | +1.90 | -19.59 | -16.20 | -7.15 | -11.89 |
>
> **Table R4 shows that performance degrades gracefully under reduced lead availability**, indicating that LVCG remains robust when only partial lead information is available.
>
> ---
>
> ### W3: Fairness of comparison with HeartLang
> We appreciate this important concern. **We follow the same model configuration and hyperparameter settings as HeartLang to ensure a fair comparison.**
>
> We acknowledge that the current manuscript does not clearly specify these details. **We will revise the paper to explicitly document all experimental settings**, including pretraining scale, model size, and evaluation protocol, to improve transparency and reproducibility.
>
> ---
>
> ### W4: Potential confounding in non-cardiac evaluation
> We thank the reviewer for pointing this out. **We agree that there may exist potential confounding factors**, as both pretraining and evaluation are conducted on related patient populations.
>
> However, we note that the evaluation tasks are defined on different label spaces and clinical objectives.
> **The observed transfer performance still suggests that LVCG captures generalizable representations beyond specific task supervision.**
>
> We will clarify this limitation in the revised version and discuss it more explicitly.
>
> ---
>
> ### W5: Bottleneck design
> We thank the reviewer for this important observation. While the bottleneck design improves cross-dataset generalization, it introduces a trade-off with in-distribution performance.
>
> **Our hypothesis is that the bottleneck enforces stronger invariance by removing redundant or lead-specific information**, which benefits transfer but may discard subtle details useful for in-distribution tasks.
>
> We will further analyze this trade-off and include a more detailed discussion in the revised manuscript.
>
> ---
>
> ### W6: Reconstruction metrics (PSNR vs alternatives)
> We thank the reviewer for this important question. We initially reported PSNR for consistency with prior work, but we acknowledge that it is not well-suited to ECG because it is highly sensitive to the predefined signal range, which is often ill-defined or dataset-dependent and therefore does not reliably reflect perceptual or physiological quality.
>
> In contrast, clinically meaningful ECG reconstruction quality is better characterized by morphology preservation rather than global amplitude fidelity. Therefore, we evaluate reconstruction quality using morphology-aware metrics that directly reflect clinically relevant features.
>
> ### Table R5: Reconstruction Metrics (PTB dataset)
> | Setting | Model | R-peak err (ms) ↓ | QRS dur err (ms) ↓ | QT err (ms) ↓ | PR err (ms) ↓ | ST-level err (mV) ↓ |
> |---|---|---:|---:|---:|---:|---:|
> | (3, 12)| NefNet | 366.21 | 33.74 | 116.27 | 22.81 | 0.23 |
> | (3, 12) | LVCG | 333.39 | 23.85 | 91.71 | 20.48 | 0.20 |
> | (5, 12)| NefNet | 782.84 | 24.63 | 88.63 | 25.22 | 0.23 |
> | (5, 12) | LVCG | 374.04 | 23.94 | 91.06 | 22.81 | 0.21 |
>
> **Table R5 shows that LVCG achieves improved morphology-related metrics, indicating better preservation of clinically relevant signal characteristics.**

---

> > ### Author Rebuttal · Reviewer_MkJU · 2026-04-03
> >
> > My concerns have been adequately addressed. Hence, I will maintain my score.

---

> > > ### Author Response · Authors · 2026-04-04
> > >
> > > We thank the reviewer for the positive feedback and are glad that our rebuttal has addressed the concerns. We appreciate the careful evaluation and consideration.

---

### Official Review · Reviewer_CP6p · 2026-03-11

**Soundness:** 3
**Presentation:** 3
**Significance:** 3
**Originality:** 3
**Overall Recommendation:** 4
**Confidence:** 4

**Summary:**

This paper proposes LVCG, a self-supervised framework for learning ECG representations in a latent vectorcardiogram (VCG) space rather than directly in the 12-lead ECG signal space. The key insight is that multi-lead ECGs are linear projections of an underlying 3D cardiac electrical field, so learning representations in this physically grounded 3D space should reduce redundancy and improve generalization. The framework uses a fixed lead direction matrix to "lift" ECG patches to VCG space via regularized least-squares, encodes beat-level VCG patches into compact tokens via a bottleneck encoder, models inter-beat dynamics with a GRU-based temporal module, and trains via masked multi-lead reconstruction. Pretrained on MIMIC-IV-ECG (~800K recordings), LVCG is evaluated on linear probing classification (PTB-XL, CPSC 2018, CSN), multi-lead reconstruction (PTB, CPSC 2018), and non-cardiac condition detection, showing strong performance particularly in low-label and cross-dataset settings.

**Compliance With Llm Reviewing Policy:**

Affirmed.

**Key Questions For Authors:**

1. Are the 1% and 10% results in Table 1 computed from a single random subset of labels, or averaged over multiple random draws? If a single draw, can you report results over (say) 5 random subsets with standard deviations? Additionally, which baseline numbers were reimplemented by you versus taken directly from prior publications (e.g., HeartLang)? Were the same label subsets used across all methods? This is critical for trusting the low-label comparisons where LVCG shows its largest advantages.

2. Can you provide an ablation comparing the temporal module with teacher forcing (feeding real encoder tokens) versus the current autoregressive setup (feeding predicted tokens)? The current design is not clearly motivated, and understanding its impact would clarify whether the autoregressive formulation is learning meaningful beat-to-beat dynamics or primarily providing regularization.

3. How does LVCG handle electrode misplacement in practice? The fixed lead direction matrix assumes standard electrode positions, but precordial electrode shifts of 1-2 intercostal spaces are common in clinical practice (Kania et al., 2014). How does this affect the lifted VCG and downstream performance? Even a simple sensitivity analysis with perturbed direction vectors would be informative and would directly support the generalization claims.

4. In Table 4, the shuffled direction matrix outperforms the learnable transformation on several tasks despite using an incorrect geometry. Do you have an explanation? One hypothesis is that a fixed (even incorrect) geometric constraint still provides useful regularization, while a learnable transformation can overfit. Understanding this would strengthen the ablation narrative.

**Limitations:**

The authors acknowledge the single-dipole approximation as a limitation. However, they do not discuss (1) the assumption of correct electrode placement baked into the fixed lead direction matrix, (2) the lack of testing across different lead systems or device types, or (3) the statistical reliability of low-label results with potentially different subsampling across methods. The comparison also excludes SOTA ECG-text multimodal methods (MERL, MELP) that represent a stronger reference point for current ECG representation learning (and to be really sure that the SOTA envelope is being pushed forward, it seems clear that ECG-text data is needed, and it can’t be done with just ECG alone). The limitations section should be more thorough.

**Strengths And Weaknesses:**

Strengths:
- The core idea of learning representations in VCG space rather than ECG signal space is well-motivated and, to my knowledge, novel for self-supervised representation learning. While VCG has been used for lead synthesis (Nef-Net), framing it as an inductive bias for learning view-invariant representations is a distinct contribution. The mathematical formulation is clean: the lift/project operations via Tikhonov-regularized pseudoinverse (Eq. 3-4) and the identifiability statement (Appendix A.1) are well-grounded.
- The VCG geometry ablation (Table 4) is the most convincing result in the paper (and without space constraints, Table 10 would also be great to have in the main paper to show that this finding is robust across dataset size regimes). The large performance drops from removing the VCG transformation, making it learnable, or shuffling the direction matrix clearly demonstrate that the fixed physically-grounded geometry is doing real work. This is not just another architecture; the inductive bias matters.
- Lead sparsity robustness (Figure 5) is practically valuable. AUROC curves remain nearly flat from 12 leads down to 3, which has real clinical implications for wearable/patch devices with limited lead configurations.  Would also be useful to fit this in the main paper somehow, if there’s space, as it’s a great result.  (How were the subsets of leads chosen? Randomly each time, or was there some logic to which leads?)
- The evaluation is broad: 6 classification datasets at 3 label fractions, multi-lead reconstruction on 2 datasets, and 3 non-cardiac tasks. Code is released.


Weaknesses:
- The paper overstates robustness to electrode placement variability and generalization across lead systems. The VCG transformation uses a fixed lead direction matrix A (Table 7) that assumes standard electrode positions. If electrodes are misplaced (which is common in clinical practice, as noted by Kania et al., 2014, which the paper itself cites), the input ECG is already corrupted before the lift, so the recovered VCG will be incorrect. The claim should be about reducing inter-lead redundancy, not robustness to acquisition geometry. Furthermore, cross-device and cross-lead-system generalization is not actually tested. All evaluation datasets use standard 12-lead ECGs; a convincing test would involve training on one lead system and evaluating on another.
- The performance framing is misleading. At 100% labels, HeartLang outperforms LVCG on most PTB-XL tasks by a substantial margin (Super: 87.52 vs 80.13, Sub: 88.91 vs 79.19, Rhythm: 90.34 vs 83.94). LVCG's strength is specifically in low-label and cross-dataset transfer (CPSC, CSN), which is a real and valuable advantage but should be stated more precisely. The paper also excludes ECG-text multimodal baselines (MERL, MELP) that leverage paired clinical reports and generally achieve stronger performance. This scoping decision is acknowledged and reasonable for a pure SSL paper, but it means the strong claims about general superiority do not hold against the actual state of the art in ECG representation learning.  Perhaps out of scope for this paper, but would be super cool to see the combination of this with multimodal ECG-text framing in future work.
- Statistical reliability of the low-label results is a significant concern. The 1% and 10% experiments involve subsampling the training set. Is this a single random subset, or averaged over multiple draws? If a single draw, the results at 1% (sometimes only ~200 samples) could change substantially with a different random split. The paper reports no confidence intervals or standard deviations across runs. This matters most where LVCG shows its largest advantages. Additionally, the paper notes that "several baseline results for linear probing classification are obtained from the original HeartLang paper" rather than reimplemented. It is unclear whether the same label subsets were used for 1% and 10% comparisons, which would be necessary for a valid head-to-head comparison. Clarifying which baselines were reimplemented versus taken from prior work, and whether identical data subsets were used, is important for reproducibility.
- The temporal module design choices are not well-motivated. Why unroll autoregressively on predicted tokens rather than real encoder tokens (teacher forcing)? What does the dynamic embedding capture that the structural embedding (mean-pooled beat tokens) does not? The ablation (Table 5, w/o Temporal) shows a drop, but the paper doesn't analyze what the temporal module is learning. For instance, is it primarily capturing heart rate variability patterns, or beat-to-beat morphological evolution?
- The multi-lead reconstruction evaluation (Table 2) and the qualitative visualization (Figure 3) are hard to interpret. The PCA embedding density plot compares two methods that produce embeddings in different spaces, making the comparison not straightforward. For reconstruction quality, it would be more informative to evaluate preservation of clinically meaningful morphological features (ST segment level, QRS duration, P-wave timing, interval widths) rather than aggregate MSE/MAE, which can be dominated by amplitude scaling or baseline wander. The appendix reconstruction plots (Figures 6-8) also appear to show substantial low-frequency drift in the ground truth signals (especially in the beginning of the traces), raising questions about whether preprocessing adequately removes baseline wander artifacts.
- The paper frames multi-lead ECG learning as either treating leads as independent channels or operating in VCG space, but there is a middle ground not discussed. Methods like ST-MEM mask in both lead and time dimensions simultaneously, partially addressing cross-lead structure without requiring a fixed physical model. Could the LVCG framework extend to joint lead-and-time masking, where different subsets of leads are visible at different beats? This seems like a natural extension that could further strengthen the approach.
- Minor: some tables do not include the metric (AUC) in their captions, making them not self-contained.
- As a nice-to-have, evaluation on newer large-scale structural heart disease benchmarks (e.g., EchoNext) would strengthen the generalization claims, given the paper's emphasis on broad transfer.

---

> ### Author Rebuttal · Authors · 2026-03-31
>
> ## Rebuttal to Reviewer CP6p
>
> ### W1 & Q3: Electrode placement variability and generalization
> We thank the reviewer for raising this important concern. **We address the simulation study on electrode placement perturbations, with full results in Table R8, in our response to Reviewer paWM (Q1).**
>
> ---
>
> ### W2: Performance framing
> Thank you for this valuable suggestion. **We agree that our current wording may overstate the general superiority of LVCG.**
>
> We will revise the manuscript to more precisely reflect that:
> - LVCG’s primary advantage lies in *low-label* and *cross-domain generalization* settings
> - It does not consistently outperform all baselines in the full-label regime
>
> ---
>
> ### W3 & Q1: Statistical reliability of low-label results
> We appreciate this important concern. To ensure statistical reliability, **we re-run all probing experiments using 5 different random seeds** for each label setting.
>
> ### Table R2: Multi-seed AUC Results (%)
>
> | Model | PTBXL-Super |  |  | PTBXL-Sub |  |  | PTBXL-Form |  |  | PTBXL-Rhythm |  |  | ICBEB |  |  | Chapman |  |  |
> |---|---|---|---|---|---|---|---|---|---|---|---|---|---|---|---|---|---|---|
> |  | 1% | 10% | 100% | 1% | 10% | 100% | 1% | 10% | 100% | 1% | 10% | 100% | 1% | 10% | 100% | 1% | 10% | 100% |
> | heartlang | 76.61 ± 0.28 | 81.68 ± 0.10 | 85.16 ± 0.07 | 66.76 ± 1.04 | 73.89 ± 0.88 | 79.93 ± 0.99 | 49.95 ± 5.22 | 66.31 ± 0.85 | 76.41 ± 0.24 | 56.50 ± 1.14 | 72.65 ± 0.76 | 85.38 ± 0.24 | 61.34 ± 0.70 | 70.71 ± 0.66 | 83.59 ± 0.18 | 62.62 ± 6.34 | 72.13 ± 0.63 | 83.43 ± 0.11 |
> | lvcg | 76.14 ± 0.13 | 80.08 ± 0.02 | 82.99 ± 0.20 | 69.92 ± 0.38 | 74.49 ± 0.43 | 80.07 ± 0.19 | 51.66 ± 3.47 | 60.95 ± 0.80 | 73.23 ± 0.25 | 72.27 ± 0.92 | 80.79 ± 0.46 | 85.96 ± 0.17 | 71.44 ± 0.15 | 79.38 ± 0.22 | 85.37 ± 0.35 | 64.48 ± 0.46 | 74.33 ± 0.15 | 84.83 ± 0.13 |
>
> **Table R2 shows that LVCG achieves stable performance with low variance across seeds, indicating that the improvements are robust rather than due to favorable sampling.**
>
> ---
>
> ### W4 & Q2: Temporal module design (teacher forcing)
> We thank the reviewer for this insightful suggestion. We conduct an additional ablation comparing the current autoregressive formulation with a teacher-forcing variant.
>
> ### Table R3: Temporal Module Ablation (AUROC @ 1%)
> | Variant | PTBXL-Super | PTBXL-Sub | ICBEB | Chapman |
> |---|---:|---:|---:|---:|
> | teacher forcing | 74.29% ± 0.22% | 68.36% ± 0.49% | 70.73% ± 0.12% | 60.37% ± 0.91% |
> | teacher forcing (trained)| 71.62% ± 0.08% | 65.02% ± 1.04% | 67.38% ± 0.62% | 54.35% ± 1.29% |
> | original LVCG (autoregressive) | 76.85% ± 0.15% | 70.23% ± 0.24% | 71.19% ± 0.32% | 64.42% ± 0.45% |
>
> **Table R3 shows that the autoregressive formulation consistently achieves the best performance**, suggesting that modeling dependencies over predicted latent tokens provides a useful regularization effect and encourages learning more robust temporal dynamics.
>
> ---
>
> ### W5: Reconstruction evaluation
> We agree that standard metrics such as MSE/MAE may not fully capture clinically meaningful signal quality.
> **We provide additional reconstruction evaluation metrics (see our response to Reviewer MkJU, W6, and Table R5)**, which better reflect morphology preservation and physiological consistency.
>
> ---
>
> ### W6: Joint lead-time masking
> Thank you for this constructive suggestion. **We agree that extending LVCG to joint lead-time masking is a promising direction**, and we will explore this in future work.
>
> ---
>
> ### W7: Metric clarity
> We appreciate this comment. **We will ensure that all tables clearly specify the evaluation metric (AUROC)** in the camera-ready version.
>
> ---
>
> ### W8: Additional benchmarks
> Thank you for this suggestion. **We agree that evaluating on newer large-scale benchmarks such as EchoNext would further strengthen the paper**, and we will include this in future work.
>
> ---
>
> ### Q4: Learnable vs fixed transformation
> This is an insightful observation. **Learning the correct electrode geometry purely from data is challenging**, as it requires recovering a physically constrained mapping under noise and limited supervision.
>
> In contrast, the ECG-to-VCG transformation is a **well-established physical relationship**, and incorporating it as a fixed prior provides a strong inductive bias and prevents overfitting.
>
> **This explains why even incorrect but fixed transformations can outperform learnable ones**, as they still impose useful structural constraints, whereas learnable mappings may drift away from physically meaningful solutions.

---

> > ### Author Rebuttal · Reviewer_CP6p · 2026-04-03
> >
> > We thank the authors for the responsive and thorough rebuttal. The new experiments directly address our primary concerns.
> >
> > The multi-seed results (Table R2) resolve our concern about statistical reliability. The low variance across 5 seeds confirms that LVCG's advantages in the low-label regime are robust to random sampling rather than artifacts of a favorable split. This was our most significant concern and the evidence is convincing.
> >
> > The teacher-forcing ablation (Table R3) is also informative and directly answers Q2. The consistent advantage of the autoregressive formulation supports the interpretation that modeling dependencies over predicted tokens provides meaningful regularization for learning temporal dynamics. We appreciate the authors running this comparison.
> >
> > The electrode perturbation analysis (Table R8, from Reviewer paWM's response) provides useful evidence that LVCG is reasonably robust to moderate geometric distortions, addressing Q3. The authors' agreement to temper the robustness claims and acknowledge the simulation-based nature of this evaluation is appropriate.
> >
> > The agreement to revise the performance framing (W2) to more precisely reflect LVCG's strengths in low-label and cross-domain settings rather than claiming general superiority is welcome. The morphology-aware reconstruction metrics (Table R5) are a useful addition that better characterize reconstruction quality than aggregate MSE/MAE.
> >
> > Regarding Q4 (learnable vs fixed), the explanation that fixed geometric constraints provide regularization even when incorrect is reasonable and aligns with our original hypothesis, though we would have appreciated a more detailed analysis (e.g., examining what the learnable transformation converges to). This is a minor point.
> >
> > We note that additional benchmarks (W8) and joint lead-time masking (W6) are deferred to future work, which is understandable given rebuttal constraints. We continue to believe these would strengthen the paper and encourage the authors to pursue them.
> >
> > Overall, the rebuttal addresses our key concerns with concrete new experiments. We are happy to adjust our score accordingly.

---

> > > ### Author Response · Authors · 2026-04-04
> > >
> > > We sincerely thank the reviewer for the thoughtful and constructive feedback. We are glad that the additional experiments have addressed the key concerns and improved the clarity and credibility of our work.
> > >
> > > We particularly appreciate the insightful suggestions on robustness, temporal modeling, and evaluation design, which have helped strengthen the paper and will also guide our future work.
> > >
> > > Thank you again for the positive assessment and for considering an updated score.

---

### Official Review · Reviewer_Tz6h · 2026-03-12

**Soundness:** 2
**Presentation:** 2
**Significance:** 1
**Originality:** 2
**Overall Recommendation:** 2
**Confidence:** 4

**Summary:**

This study addresses the inherent issue of redundancy introduced by representation learning in the ECG signal space. The authors propose a novel learning framework, termed LVCG, which is designed to directly learn a unified latent representation of cardiac electrical activity in the VCG space. By focusing on learning view-invariant latent VCG representations—rather than capturing lead-specific artifacts—the LVCG framework effectively minimizes redundancy and enhances model generalization.

**Compliance With Llm Reviewing Policy:**

Affirmed.

**Key Questions For Authors:**

1.  Although PSNR originated from image processing, it is still a widely accepted fidelity metric in signal reconstruction. Why is the reconstruction by LVCG still considered “better physiological consistency” when the PSNR is relatively low?
2.Since the single dipole assumption ignores near-field effects and non-dipolar components, does the forced compression of 12-lead ECG into a 3-dimensional VCG space risk losing critical diagnostic information?
3. If the geometric distribution of the visible leads is poor (e.g., all limb leads without chest leads), could the model’s “Lift” operation in the VCG space lead to severe numerical instability?

**Limitations:**

Yes.

**Strengths And Weaknesses:**

Strengths：
1.This study develops a general representation learning framework in the latent VCG space, offering a physically interpretable and verifiable alternative to direct ECG space modeling.
2.This study effectively captures the evolutionary dynamics of cardiac activity in the latent VCG space by integrating beat-level structural features with inter-beat dynamics.

Weakness
1.The experimental results indicate that the LVCG model underperforms compared to the baseline methods. As shown in Table 1, the AUC scores achieved by LVCG are consistently and significantly lower than those of the other models across all evaluation tasks.
2.The evaluation relies solely on the AUC metric. To provide a more comprehensive and robust assessment of model performance, it is recommended to incorporate additional widely-used metrics, such as Accuracy and F1-score. This would offer deeper insights into the models’ behavior across different performance dimensions.
3.The clarity of the methodological exposition, particularly the illustration in Figure 1, could be enhanced. The current presentation makes the proposed framework somewhat difficult to follow. Additionally, Figure 1 contains minor formatting issues; for instance, a portion of the heart diagram is partially cropped or obscured, which detracts from the visual clarity.
4.Since a standard 12-lead ECG can be transformed into a VCG through established linear transformations, the necessity of the authors’ proposed learning-based mapping into the latent VCG space warrant further justification.

---

> ### Author Rebuttal · Authors · 2026-03-31
>
> ## Rebuttal to Reviewer Tz6h
>
> ### W1: Performance comparison
> We thank the reviewer for the detailed evaluation. We respectfully clarify that the conclusion that LVCG “consistently and significantly underperforms” is not supported by the overall results.
>
> **In Table 1, LVCG achieves the best performance in 9 settings across label regimes, the highest among all methods**, indicating that it does not consistently underperform. The observed discrepancy likely arises from focusing on the full-label regime, where some baselines achieve slightly higher peak performance.
>
> Importantly, **LVCG is the only method that remains competitive across all tasks** (ECG classification, multi-lead reconstruction, and non-cardiac prediction), a property also recognized by other reviewers (CP6p, MkJU, paWM).
>
> For ECG classification, LVCG shows its strongest advantage under low-label and cross-dataset settings, which are more relevant in real-world scenarios. This reflects improved label efficiency and transferability rather than optimization for fully supervised settings.
>
> **Therefore, not achieving the best performance in every full-label setting does not weaken the contribution of LVCG, but instead reflects its focus on generalization and robustness.**
>
> We will revise the manuscript to clearly distinguish performance across regimes and moderate claims accordingly.
>
> ---
>
> ### W2: Evaluation metrics
> We appreciate the suggestion to include additional evaluation metrics. As a self-supervised learning method, we prioritize AUROC because it is threshold-independent and better reflects representation separability than F1 (which depends on a specific operating point and calibration), and because reporting AUROC matches prior ECG representation learning work for fair comparison. In addition, ECG classification is typically highly imbalanced, so accuracy is less informative.
>
> To address the reviewer’s concern, we additionally compute the F1-score on the same setting in Table R1.
>
> ### Table R1: F1 Results (%)
>
> | Model | PTBXL-Super |  |  | PTBXL-Sub |  |  | PTBXL-Form |  |  | PTBXL-Rhythm |  |  | ICBEB |  |  | Chapman |  |  |
> |---|---|---|---|---|---|---|---|---|---|---|---|---|---|---|---|---|---|---|
> |  | 1% | 10% | 100% | 1% | 10% | 100% | 1% | 10% | 100% | 1% | 10% | 100% | 1% | 10% | 100% | 1% | 10% | 100% |
> | heartlang | 52.87 | 59.52 | 63.39 | 16.04 | 22.22 | 30.20 | 12.14 | 15.92 | 20.51 | 16.93 | 31.26 | 43.08 | 26.99 | 38.52 | 52.91 | 21.37 | 38.09 | 57.97 |
> | lvcg | 53.73 | 57.03 | 60.61 | 17.44 | 22.14 | 25.41 | 11.56 | 16.15 | 21.87 | 18.97 | 24.87 | 31.35 | 33.11 | 43.14 | 51.00 | 31.11 | 45.08 | 52.48 |
>
> **Table R1 shows that F1 results are consistent with AUROC trends, supporting the robustness of our conclusions.**
>
>
> ---
>
> ### W3: Clarity of presentation
> Thank you for pointing this out. Based on the description, this concern mainly relates to visualization clarity rather than the conceptual framework. Figure 1 is intended as a high-level conceptual illustration, while the detailed method pipeline is presented in Figure 2.
>
> We will revise the figures in the camera-ready version to improve clarity, including refining the pipeline visualization and ensuring that all graphical elements are fully visible and clearly presented.
>
> ---
>
> ### W4: Necessity of the VCG transformation
> We would like to clarify that the ECG-to-VCG transformation in LVCG is **non-learnable** and defined by a fixed linear mapping, as described in Sec. 3.2 (“VCG Space Transformation”). Therefore, our method does not introduce an additional learnable mapping from ECG to VCG.
> We will make this distinction more explicit in the revised version to avoid confusion.
>
> ---
>
> ### Q1: PSNR vs physiological consistency
> We thank the reviewer. **Please see our response to Reviewer MkJU (W6) and Table R5** for the full discussion and morphology-aware metrics. **We will move PSNR to the appendix and foreground morphology-based metrics in the main text.**
>
> ### Q2: Does compression to VCG space lose critical diagnostic information?
> From an empirical perspective, LVCG achieves competitive performance on both cardiac and non-cardiac classification tasks, indicating that critical diagnostic information is preserved in the learned representation.
>
> Multi-lead reconstruction (Table R5; Reviewer MkJU, W6) also shows strong recovery of morphology-sensitive features (e.g., ST/interval metrics), supporting retention of clinically relevant information under compression.
>
> ---
>
> ### Q3: Stability under poor lead geometry (e.g., only limb leads)
> We thank the reviewer. **Reduced-lead robustness is reported in our response to Reviewer MkJU (W2; Table R4), and electrode-geometry perturbations in our response to Reviewer paWM (Q1; Table R8).** Together, these results support that LVCG remains stable under imperfect lead availability and moderate placement variation.

---

> > ### Author Rebuttal · Reviewer_Tz6h · 2026-04-03
> >
> > 1. I find the authors’ response regarding the LVCG model’s performance unconvincing. HeartLang also demonstrates strong overall performance, significantly outperforming LVCG on several tasks, including PTBXL-Super, PTBXL-Sub, and PTBXL-Form. Moreover, in the supplementary Table R1, HeartLang’s F1 scores substantially exceed those of LVCG, with discrepancies reaching approximately 12 percentage points in the worst case.
> >
> > 2. I understand that the VCG in LVCG can be derived from ECG through a fixed linear transformation. Given that this is merely a fixed linear mapping, ECG and VCG essentially reside in the same space—raising the question of what meaningful advantage this processing offers.
> >
> > 3. The authors claim that critical diagnostic information is preserved in the learned representations when transforming ECG to VCG. However, their results, particularly the F1 scores, provide no substantiation for this assertion.
> >
> > I maintain my current assessment.

---

> > > ### Author Response · Authors · 2026-04-04
> > >
> > > We sincerely thank the reviewer for this thoughtful feedback. We believe that the **remaining concerns may partly reflect limitations in how we presented the results and articulated our contributions**, and we are grateful for the opportunity to clarify these points more carefully. These comments will help us significantly improve the paper in the revised manuscript.
> > >
> > > We understand that the main concerns focus on two aspects: (1) the **performance of LVCG**, and (2) the practical **advantages of modeling in the VCG space**, including whether it leads to **information loss**.
> > >
> > > ### 1. Performance
> > >
> > > We systematically summarize the performance of LVCG across the three tasks studied in this paper.
> > >
> > > | Task | Main finding | Evidence |
> > > |------|-------------------|----------|
> > > | Linear probing ECG classification | LVCG shows clear advantages in the 1% and 10% label settings | Table 1, Table R1 |
> > > | Non-cardiac condition prediction | LVCG shows clear advantages | Table 5 |
> > > | Multi-view reconstruction | LVCG shows clear advantages | Table 2, Figure 3, Table R5 |
> > >
> > > We believe that the reviewer's concern is mainly centered on the linear probing ECG classification task. To clarify this more precisely, we summarize the overall comparison across different label regimes in Table R9.
> > >
> > > **Table R9. Average results for linear probing ECG classification across different label regimes.**
> > >
> > > | Metric | Label % | LVCG Avg | HeartLang Avg |
> > > |--------|--------|----------|---------------|
> > > | AUC    | 1%     | **67.98**    | 62.96         |
> > > | AUC    | 10%    | **74.35**    | 72.23         |
> > > | AUC    | 100%   | 82.74    | **82.98**     |
> > > | F1     | 1%     | **27.65**    | 24.39         |
> > > | F1     | 10%    | **34.73**    | 34.26         |
> > > | F1     | 100%   | 40.45    | **44.68**     |
> > >
> > > A clearer pattern emerges: LVCG shows its strongest advantages under **1% and 10% label settings** and in **cross-dataset settings**, particularly on the distribution-shift datasets (ICBEB / Chapman), while in full-label settings some baselines achieve higher peak performance.
> > >
> > > Overall, across the three task types, LVCG exhibits the following properties:
> > > - Clear gains in **low-label** and **cross-dataset** classification settings
> > > - Strong advantages in **non-cardiac** and **reconstruction tasks**
> > >
> > > Taken together, these results highlight the strengths of LVCG in **label-efficient learning, cross-dataset generalization, and broader task transferability**.
> > >
> > > ---
> > >
> > > ### 2. Advantages of VCG representation and information preservation
> > >
> > > **Advantages:**
> > >
> > > (1) **Generalizable representations**
> > > The VCG space provides a physically constrained low-dimensional representation, which helps remove redundant information and improves **generalization across tasks and distributions**.
> > >
> > > (2) **Computational and data efficiency**
> > > The VCG representation is 3-dimensional, compared to 12-dimensional ECG signals, leading to improved efficiency in both parameter count and computational cost (Table R10).
> > >
> > > **Table R10. Parameter count and inference time comparison across models.**
> > >
> > > | Model     | Params | Inference (B=8) |
> > > |-----------|--------|-----------------|
> > > | LVCG      | 8.29M  | 32 ms           |
> > > | HeartLang | 39.05M | 146 ms          |
> > > | ST-MEM    | 88.53M | 120.34 ms       |
> > >
> > > (3) **Robustness to missing leads**
> > > Since ECG-to-VCG is a many-to-one mapping, VCG representations are inherently more robust to missing leads. This is supported by results under lead removal and perturbation (Tables R4 and R8, and Figure 5).
> > >
> > > ---
> > >
> > > **On information loss:**
> > >
> > > We acknowledge that compressing a 12-lead ECG into a 3-dimensional VCG representation cannot preserve all information. The key question, however, is whether **diagnostically relevant information is retained**.
> > >
> > > Empirically:
> > > - LVCG achieves competitive performance on linear probing ECG classification (Table 1, Table R9)
> > > - LVCG significantly outperforms baselines on non-cardiac condition prediction (Table 5)
> > > - Table R5 shows strong preservation of morphology-sensitive features in the reconstruction task (e.g., ST-segment and interval-related metrics)
> > >
> > > These results indicate that the VCG representation **preserves sufficient diagnostic information in practice**, and in many scenarios outperforms ECG-based models with far more parameters.
> > >
> > > ---
> > >
> > > We again thank the reviewer for the thoughtful and critical feedback. These comments have helped us better **clarify the scope and positioning of LVCG**. We will further improve the presentation of results and the articulation of contributions in the revised version.

---

### Decision · Program_Chairs · 2026-04-30

**Decision:**

Accept (regular)

**Comment:**

The authors propose LVCG, a self-supervised learning framework that learns ECG representations in a physically grounded latent vectorcardiogram (VCG) space, rather than directly from multi-lead signals, to remove redundancy and enhance generalisation.

The key strengths are the novel and well-motivated use of VCG geometry as an inductive bias, the thorough multitask evaluation, and the persuasive empirical demonstrations of strong label efficiency and cross-dataset transferability (especially in low-label regimes).

The major downsides are related to a lingering concern from one reviewer about absolute performance parity with strong baselines in fully supervised scenarios and the ultimate usefulness of the fixed VCG transformation, despite numerous rebuttal comments providing clarifications and evidence of LVCG's competitiveness and unique benefits in generalisation and efficiency.

The paper received a majority of positive comments from knowledgeable reviewers of this paper, and the primary concerns regarding statistical reliability and the overall experimental framing were addressed by additional multi-seed and ablation results in the rebuttal. While one reviewer still found the performance narrative unconvincing, the general consensus is that this is a novel paper with strong results in label-efficient and cross-domain transfer learning.